# Dynamic enlargement and mobilization of lipid droplets in pluripotent cells coordinate morphogenesis during mouse peri-implantation development

King Hang Tommy Mau[1,8], Donja Karimlou[1], David Barneda[1,9], Vincent Brochard[2,3], Christophe Royer[4], Bryony Leeke[5,6], Roshni A. de Souza[1], Mélanie Pailles[2,3], Michelle Percharde [1,5,6], Shankar Srinivas [4], Alice Jouneau [2,3], Mark Christian [1,7,10] & Véronique Azuara [1✉]

Mammalian pre-implantation embryos accumulate substantial lipids, which are stored in lipid droplets (LDs). Despite the fundamental roles of lipids in many cellular functions, the significance of building-up LDs for the developing embryo remains unclear. Here we report that the accumulation and mobilization of LDs upon implantation are causal in the morphogenesis of the pluripotent epiblast and generation of the pro-amniotic cavity in mouse embryos, a critical step for all subsequent development. We show that the CIDEA protein, found abundantly in adipocytes, enhances lipid storage in blastocysts and pluripotent stem cells by promoting LD enlargement through fusion. The LD-stored lipids are mobilized into lysosomes at the onset of lumenogenesis, but without CIDEA are prematurely degraded by cytosolic lipases. Loss of lipid storage or inactivation of lipophagy leads to the aberrant formation of multiple cavities within disorganised epithelial structures. Thus, our study reveals an unexpected role for LDs in orchestrating tissue remodelling and uncovers underappreciated facets of lipid metabolism in peri-implantation development.

[1] Institute of Reproductive and Developmental Biology, Department of Metabolism, Digestion, and Reproduction, Faculty of Medicine, Imperial College London, Hammersmith Hospital, Du Cane Road, London W12 0NN, UK. [2] Université Paris-Saclay, UVSQ, INRAE, BREED, Jouy-en-Josas 78350, France. [3] École Nationale Vétérinaire d'Alfort, BREED, Maison-Alfort 94700, France. [4] Institute of Developmental & Regenerative Medicine, Department of Physiology, Anatomy and Genetics, University of Oxford, Old Road Campus, Oxford OX3 7TY, UK. [5] MRC London Institute of Medical Sciences (LMS), Hammersmith Hospital, Du Cane Road, London W12 0NN, UK. [6] Institute of Clinical Sciences, Faculty of Medicine, Imperial College London, Hammersmith Hospital, Du Cane Road, London W12 0NN, UK. [7] Division of Biomedical Sciences, Warwick Medical School, University of Warwick, Coventry CV4 7AL, UK. [8] Present address: Physiology and Metabolism Laboratory, The Francis Crick Institute, London NW1 1AT, UK. [9] Present address: Signalling Programme, The Babraham Institute, Cambridge CB22 3AT, UK. [10] Present address: School of Science and Technology, Nottingham Trent University, Nottingham NG11 8NS, UK. ✉email: v.azuara@imperial.ac.uk

P luripotency is a transient property of embryonic cells existing from pre- to early post-implantation development. At the blastocyst stage, pluripotent epiblast progenitors (i.e., the future embryo) polarize to form rosette-like epithelial structures enclosing a central (pro-amniotic) lumen upon implantation. This morphogenesis event is fundamental for all subsequent development and hence the success or failure of a pregnancy[1–4]. Concurrently, epiblast cells adopt distinct molecular states, which delineate the transition from naive (pre-implantation) to primed (post-implantation) pluripotency[2,5]. While these molecular signatures are thought to prime or capacitate cells for differentiation, associated metabolic switches have long been regarded as survival adaptations prior to the establishment of placental exchange during embryo implantation[6,7].

Recent studies suggest, however, that metabolism can play active roles in regulating cell fate transitions during early embryogenesis[8–12]. Among cellular metabolites, lipids contribute to membrane and organelle homeostasis[13,14], energy metabolism[15] and signalling pathways[16], which are all tightly regulated and essential for developmental progression. In eukaryotic cells, lipids can be esterified and stored as inert neutral lipids in lipid droplets (LDs)[17]. Besides their neutral lipid core, LDs are made up of a phospholipid monolayer, which is decorated by various proteins. These proteins regulate lipid homeostasis, for example, the activation of lipases on the surface of LDs triggers the mobilization of stored lipids in response to cellular metabolic requirements[18]. In addition, LD-surface factors promote functional interactions between LDs and other organelles to facilitate intracellular trafficking of lipid storage[19,20]. During pre-implantation development, early mammalian embryos have been reported to actively uptake, synthesize and accumulate lipids as LDs[21–23]. In contrast, only scarce LDs are found in early post-implantation embryonic tissues[24], implying active mobilization of stored lipids during the transition from naive to primed pluripotent states. Yet the mechanisms underlying the dynamics of LDs and their biological importance have remained a mystery.

In this study, we uncover how intrinsic LD-associated mechanisms fundamentally link lipid metabolism to the control of morphogenesis in mouse peri-implantation embryos. Using three-dimensional (3D) cultures of embryonic stem cells (ESCs)[1], in combination with embryo studies, we find that the sequential storage and mobilization of lipids, mediated by LDs, coordinate the formation of an apical lumen in epiblast rosettes. Mechanistically, we show that pre-implantation blastocysts and pluripotent ESCs share the ability to transiently fuse and enlarge LDs under the tight control of the LD-surface protein CIDEA. Critically, CIDEA-mediated LD enlargement confers protection against promiscuous degradation by cytosolic lipases upon cell polarization. This permits the hydrolysis of stored lipids into lysosomes at the onset of lumen formation in a time-dependent manner. Abrogating CIDEA's function or lipophagy-dependent mobilization of LDs impairs lumenogenesis, demonstrating the functional significance of lipid storage during peri-implantation development.

## Results

**Pre-implantation blastocysts and derived ESCs share the ability to fuse and enlarge LDs**. LDs are evolutionary-conserved cellular organelles that control the storage and usage of lipids from bacteria to eukaryotic cells[17]. To understand how stored lipids accumulate in pre-implantation embryos, we examined the number, size, and behaviour of LDs at different stages of development. For this, 4–8 cell embryos (harvested ~68 h post-hCG injection) were cultured to morula (E2.5; ~74 h post hCG) and blastocyst (E3.5; ~98 h post hCG) stages in KSOMaa medium,

and LDs were visualised by neutral lipid staining (BODIPY 493/503) and confocal imaging (Fig. 1a). In pre-compacted morulae, we confirmed that LDs were numerous, small, and scattered, in agreement with previous reports using label-free methods[24,25]. While morulae developed into cavitating embryos, small LDs tended to be clustered and subsequently fused as evidenced by the detection of single and rounded BODIPY-labelled entities in fully expanded blastocysts (see zoomed views, Fig. 1a). This was accompanied by a decline in the total number of individual LDs and concomitant increase in their size as monitored under constant culture conditions (Fig. 1b, c). Together, these results indicate that mouse early embryos intrinsically acquire the ability to fuse LDs during the transition from the morula to the blastocyst stage.

Interestingly, the ability to fuse and enlarge LDs was preserved in blastocyst-derived ESCs cultured in the presence of knockout serum replacement (KSR) or AlbuMAX[26] as fatty acid sources (Fig. 2a; Supplementary Fig. 1a, b). In contrast, mouse embryonic fibroblasts (MEFs) exhibited small LDs that did not fuse in either culture conditions. Time-lapse imaging further revealed that LD enlargement in ESCs operates via a tightly regulated process with the pairing and docking of adjacent LDs that slowly fused through transfer of lipids in the course of ~4 h (Fig. 2b; Supplementary Movie 1). While LDs can rapidly coalesce, a slow process of fusion is characteristic of the action of cell death-inducing DFF45-like effector (CIDE) proteins at LD-LD contact sites as previously studied in adipocytes[27,28]. To determine whether the same LD-surface factors regulate the behaviour of LDs in pluripotent cells, we screened in ESCs and MEFs for the expression of *Cide* gene family members (*Cide-a, -b, -c*) along with perilipin factors (*Plin1-5*), which have also been implicated in the regulation of LD growth[20,29]. While all genes tested could be detected in both cell types apart from *Plin1* in ESCs, the *Cidea* transcript stood out as being highly expressed in ESCs relative to MEFs and further induced under lipid-rich conditions (Fig. 2c; Supplementary Fig. 1c, d), strictly correlating with maximum LD enlargement (Supplementary Fig. 1a, c, e, f).

**The CIDEA protein is required for the full process of LD fusion in pluripotent cells**. To test whether CIDEA induction is conducive to LD fusion in ESCs, we ectopically expressed a V5-tagged *Cidea* construct in these cells (Supplementary Fig. 2a, b), and validated the correct localisation of V5-CIDEA protein along the surface of enlarged LDs and at LD-LD contact sites upon LD fusion conditions (Fig. 2d and Supplementary Fig. 2c). As anticipated, overexpressing (OE) CIDEA in ESCs was sufficient to double the average size of LDs, whilst reducing their number relative to controls (Fig. 2e, f). Conversely, deleting the *Cidea* locus using CRISPR-Cas9 completely blocked the occurrence of LD fusion in knockout (KO) ESCs, revealing CIDEA as an essential factor in regulating this process (see Supplementary Fig. 2d–f and Supplementary Fig. 3a). As an important control, and to further infer a direct role for CIDEA, we confirmed that manipulating the expression of this factor alone was sufficient to alter the behaviour of LDs without affecting the undifferentiated status or proliferation rate of both CIDEA OE and KO ESCs under self-renewing culture conditions (see full analysis in Supplementary Fig. 2g–l). Moreover, we demonstrated that LD fusion could be fully rescued in KO cells by exogenously expressing a wild-type (WT) *Cidea* as opposed to point-mutated constructs[27], where the ability of CIDEA to stabilise LD pairs (R47E) or facilitate the transfer of lipids (R171E) between LDs was abrogated (Supplementary Fig. 3b, c). While CIDEA KO^WT ESCs regained the ability to enlarge LDs, CIDEA KO^Empty, CIDEA KO^R47E and CIDEA KO^R171E cells showed an accumulation of

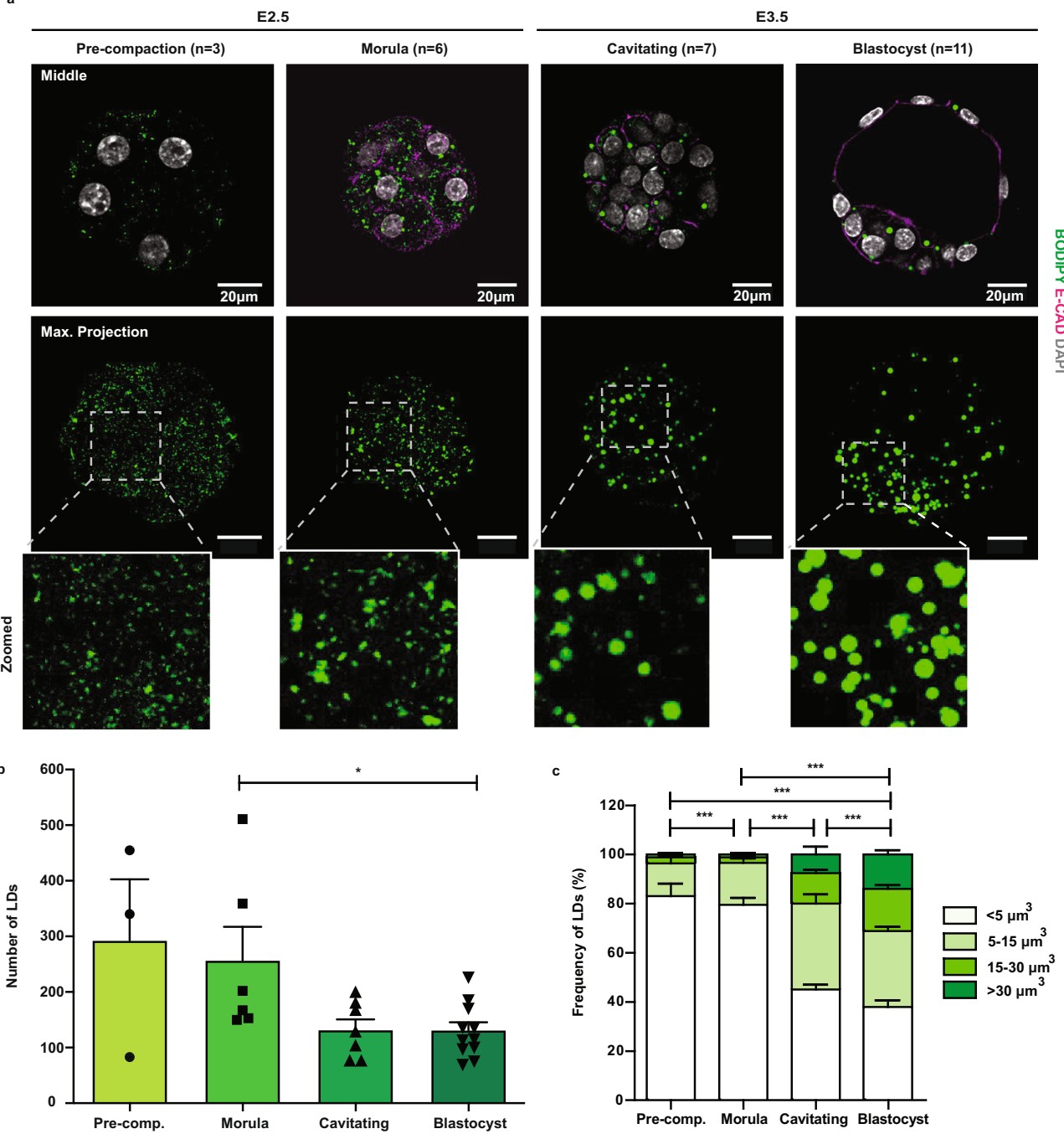

**Fig. 1 The ability to fuse and enlarge lipid droplets is acquired by the mouse embryo at the morula-to-blastocyst transition. a** Images of BODIPY 493/503-stained lipid droplets (LDs) in in vitro pre-implantation embryos cultured from 4 to 8 cell stages (~68 h post-hCG injection) in KSOMaa medium. Adherens junctions were labelled by immunostaining against E-CADHERIN ($n = 3$–11 embryos). Squares indicate magnified regions, showing LD morphology and enlargement. E, embryonic day of development. E2.5, ~74 h post-hCG injection; E3.5, ~98 h post-hCG injection. Quantification of **b** total number of LDs per embryo and **c** volume of LDs in embryos from (**a**). Each symbol represents one embryo. Error bars, means ± s.e.m.; two-sided Mann–Whitney U test; *$p < 0.05$; ***$p < 0.001$. Exact $p$-values: **b** $p = 0.0237$, and **c** pre-comp. vs morula, $p = 0.0004$; pre-comp. vs blastocyst, $p = 3.9 \times 10^{-64}$; morula vs cavitating, $p = 4 \times 10^{-43}$; morula vs blastocyst, $p = 1.3 \times 10^{-60}$; cavitating vs blastocyst, $p = 4.2 \times 10^{-8}$. Source data are provided as a Source data file.

small LDs with different degrees of clustering regardless of culture conditions (Fig. 2g; Supplementary Fig. 3d). Hence, our results demonstrate that CIDEA is strictly required to fulfil the full process of LD enlargement in ESCs, potentially to optimise lipid storage.

In vivo, the *Cidea* transcript could be readily detected at higher levels than *Cideb* and *Cidec* as seen in ESCs (Supplementary

Fig. 1g). CIDEA protein was also accumulated at the surface of enlarged LDs as exemplified in the inner cell mass of the E3.5 blastocyst (Fig. 2h; see also Supplementary Fig. 1h for additional validation). Mirroring the onset of LD enlargement (Fig. 1a), we verified using available data[30] that *Cidea* expression was specifically upregulated at the morula-to-blastocyst stage transition (Fig. 2i), pointing to a conserved function in vivo. Strikingly,

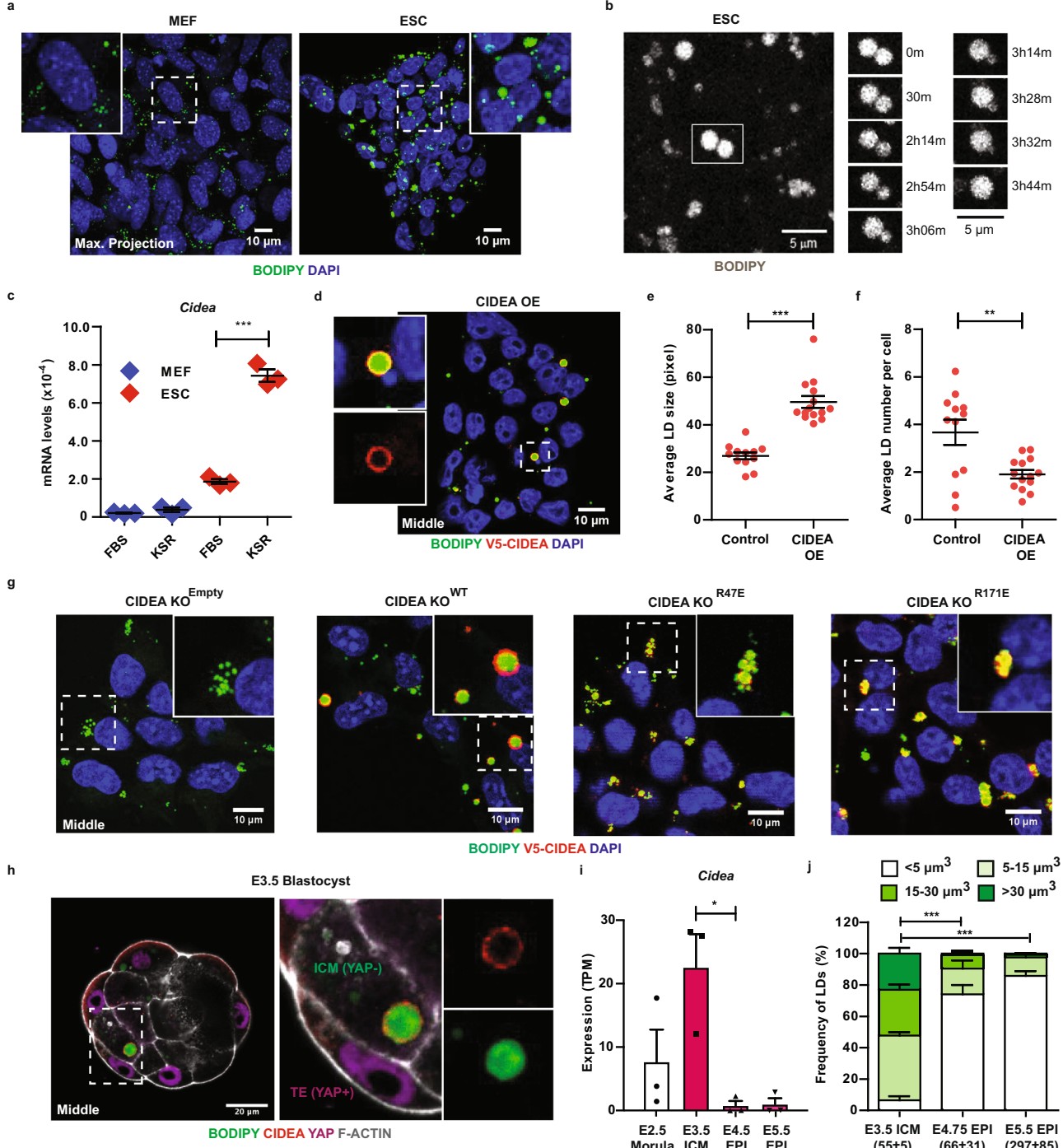

however, the *Cidea* transcript sharply declined in the developing epiblast from E4.5 onwards (Fig. 2i; Supplementary Fig. 1g), prompting us to extend our analysis of LDs to peri-implantation times. Using a lineage-specific LD quantification pipeline (Supplementary Fig. 4a, b), we observed that enlarged LDs were lost in epiblast cells of the implanting blastocyst (E4.5–E4.75) and were not regained post-implantation (E5.5). Instead, LDs appeared to be smaller and more homogeneous in size, in agreement with *Cidea* silencing (Fig. 2i, j). Enlarged LDs were, however, retained in some extra-embryonic tissues including the mural trophectoderm (TE) as opposed to the polar TE and primitive endoderm (PrE) (E4.75; Supplementary Fig. 4c, d). Collectively, our results reveal lineage-specific mobilization of lipid storage, and furthermore point to an important role for

CIDEA's function in regulating the dynamics of LDs in pluripotent progenitors both in vitro and in vivo.

**Trafficking of lipid storage coordinates polarity-driven lumenogenesis.** To delineate the significance of lipid storage in the epiblast lineage, we turned to an established ESC-based 3D spheroid model[1], which closely mimics the development of the epiblast during the pre- (E3.5) to post-implantation (E5.5) transition. Spheroid formation was here induced by embedding individual ESCs into Matrigel where they can proliferate, polarise into epiblast-like rosettes, and form a single lumen at the centre of radially arranged cells under differentiation conditions (Fig. 3a, upper panel; see also 'Methods'). Importantly, we could recapitulate in this in vitro model the dynamics of LDs, which we

**Fig. 2 CIDEA promotes lipid droplet fusion in the blastocyst and naive pluripotent stem cells. a** Representative images of BODIPY 493/503-stained LDs in MEFs and E14-ESCs cultured in knockout serum replacement (KSR)-supplemented medium ($n = 3$). Squares indicate magnified regions. **b** Time-lapse images of BODIPY-stained LDs in E14-ESCs exposed to KSR showing the stepwise fusion of LDs (white square) at indicated time intervals ($n = 2$) (see also Supplementary Movie 1). **c** *Cidea* transcript expression (RT-qPCR) in MEFs (blue) and E14-ESCs (red) cultured in foetal bovine serum (FBS) and KSR-supplemented media (see also Supplementary Fig. 1a, c). Error bars, means ± s.e.m. ($n = 3$); two-sided unpaired Student's $t$-test; ***$p < 0.001$ ($p = 0.000094$). **d** Immunostaining images of V5 epitope on the surface of BODIPY-stained LDs in E14-ESCs overexpressing (OE) V5-CIDEA in KSR-supplemented medium ($n = 3$) (see also Supplementary Fig. 2a–c). Quantification of **e** average LD size (area) in pixels and **f** average LD number per cell in control (empty vector) and V5-CIDEA OE E14-ESCs cultured in KSR-supplemented medium. Error bars, means ± s.e.m. Each red dot represents analysis of one single colony. 12 and 14 control and V5-CIDEA OE E14-ESC colonies were respectively analysed across independent experiments ($n = 3$). Two-sided unpaired Student's $t$-test; **$p < 0.01$ (**f**, $p = 0.00246$); ***$p < 0.001$ (**e**, $p = 0.00000009$). **g** Representative images of BODIPY-stained LDs and V5 epitope in CIDEA knockout (KO) ESCs (clone 19.7; Supplementary Fig. 2d–f) stably transfected with control (CIDEA KO$^{Empty}$), wild-type (CIDEA KO$^{WT}$) and point-mutated (CIDEA KO$^{R47E}$ and CIDEA KO$^{R171E}$) V5-tagged *Cidea* constructs, in KSR-supplemented medium ($n = 3$) (see also Supplementary Fig. 3d for images of rescued cells in FSB-supplemented medium). **h** Representative images of BODIPY-labelled LDs and CIDEA protein detected at the surface of LDs by immunostaining using an anti-CIDEA antibody from Proteintech (13170-1-AP) in freshly harvested E3.5 blastocysts ($n = 21$). Nuclear YAP-positive cells, trophectoderm (TE); Nuclear YAP-negative cells, inner cell mass (ICM). F-ACTIN (Phalloidin) was used as cell membrane marker (see also Supplementary Fig. 1h for additional validation). **i** Expression (RNA-sequencing; E-MTAB-2958) of *Cidea* transcript in morula (E2.5; white), dissected ICM (E3.5) and EPI (E4.5 and E5.5) shown as Transcripts Per Million (TPM). Each symbol represents one biological replicate. Two-sided unpaired Student's $t$-test, *$p < 0.05$ ($p = 0.014768$). ICM, inner cell mass; EPI, epiblast. **j** Quantification of the volume (μm³) and number of LDs in the ICM/EPI of peri-implantation embryos from E3.5 ($n = 6$), E4.75 ($n = 3$) and E5.5 ($n = 5$) (see also Supplementary Fig. 4). Error bars, means ± s.e.m. ($n = 3$–$6$). Indicated in brackets are the average numbers of LDs analysed at each developmental time. Two-sided Mann–Whitney U test, ***$p < 0.001$. Exact $p$-values: E3.5 ICM vs E4.75 EPI, $p = 1 \times 10^{-30}$; E3.5 ICM vs E5.5 EPI, $p = 8.4 \times 10^{-44}$. Source data are provided as a Source data file.

observed in peri-implantation embryos in vivo (Fig. 2j). Spheroids formed by WT ESCs were seen to build up enlarged LDs upon induction of differentiation (24 h), which were subsequently mobilized (48 h) as *Cidea* expression declined and a central nascent lumen appeared (Fig. 3a–c). In contrast, CIDEA KO spheroids failed to both enlarge LDs and accumulate neutral lipids (i.e., total BODIPY intensity) at the onset of differentiation (24 h; Fig. 3c, d) as confirmed by direct quantification of LD-stored triglycerides (24 h; Supplementary Fig. 5a). Most intriguingly, we also noted that CIDEA KO spheroids frequently displayed multiple cavities instead of a single, expanding central lumen (72 h; Fig. 3a, lower panel). This defect manifested as cells transited normally from a naive to a transcriptionally primed state of pluripotency (Fig. 3b; see also our global transcriptomic analysis in Supplementary Fig. 5b–e), which has been reported to be crucial for morphogenesis[2].

To further scrutinise the phenotype of CIDEA KO spheroids, we used the Podocalyxin (PODXL) glycoprotein as an established epithelial apical marker to quantify the process of cell polarity and lumen formation by hollowing, the result of apical membrane separation[31]. Upon differentiation, PODXL was upregulated, and the protein trafficked apically in both WT and KO spheroids at similar frequency (~40% of total spheroids examined; WT $n = 122$, and KO $n = 148$) (24 h; Supplementary Fig. 6a, b). Therefore, these results exclude an early defect in the initiation of cell polarity. Upon insertion at apical membranes (48 h), PODXL promoted, via charge repulsion[32,33], the formation of a single, central lumen in most WT spheroids (62 +/− 5%; 72 h; $n = 162$). In contrast, the formation of multi-lumened or disorganised spheroids was prevalent in CIDEA KO cultures (81 +/− 7%; 72 h; $n = 137$) (Fig. 3e; see also Supplementary Fig. 6a, b). Moreover, the distribution of all polarity markers we examined (i.e., PODXL, atypical PKC complex membrane PARD3[1]; Supplementary Fig. 6a, c), the apical repositioning of the Golgi apparatus[1,34] (via GM130; Supplementary Fig. 6d) and other organelles (Supplementary Fig. 6e, f), were all altered in CIDEA KO spheroids, most likely reflecting the formation of multiple lumens within disorganised structures. Correct lumenogenesis could, however, be rescued by re-introducing a functional CIDEA protein, as opposed to point-mutated (R47E and R171E) forms in KO spheroids (Fig. 3f and Supplementary Fig. 6g), demonstrating a prime role for CIDEA and its LD fusion activity

in this process. Taken together, our results unravel an unprecedented link between LD biology and the cellular remodelling of pluripotent progenitors.

**The emergence of multiple cavities typifies abnormal morphogenesis in vivo.** The importance of CIDEA's function was also established in the remodelling epiblast in vivo. For this, we generated embryo chimaeras by injecting stably expressing LifeAct-RFP WT and CIDEA KO ESCs into E2.5 embryos, which were then transferred back into surrogate females (Fig. 3g; see also 'Methods'). WT ($n = 24$) and KO ($n = 30$) embryo chimaeras were retrieved after three days of development (~E5.5) and assessed based on their overall morphology and their degree of epiblast colonization by RFP-positive cells. Injected ESCs that lack CIDEA were still capable of contributing to a developing embryo at comparable frequency to their WT counterparts (Supplementary Fig. 6h). Focussing on highly colonized chimaeras, however, we found that 9 out of 10 CIDEA KO embryos displayed disorganised epithelium with multi-foci of PODXL deposition amongst OCT4-positive epiblast cells (Fig. 3h, i). In contrast, no control embryos ($n = 13$) exhibited multiple lumens with most of them harbouring a single, central cavity (Fig. 3i; see also Supplementary Movies 2, 3). These results validate our findings from 3D spheroids showing that, in the absence of CIDEA, the remodelling epiblast generates multiple nascent lumens in vivo. In agreement with the non-lethality of CIDEA KO mice[35], however, CIDEA KO embryo chimaeras would show an extended, single pro-amniotic cavity when examined at later developmental times (~E7.5) (Supplementary Fig. 6i). This most likely indicates that compensatory mechanisms operate independently of CIDEA expression in vivo[36,37] (see the 'Discussion' section).

**CIDEA-mediated LD fusion facilitates lumenogenesis by safeguarding lipid storage.** To gain further mechanistic insights into how CIDEA-mediated LD fusion promotes correct lumenogenesis, we next sought to compare the trafficking of LDs in WT and CIDEA KO spheroids around the time of maximum enlargement (24 h). BODIPY pulse-chase labelling combined with time-lapse imaging revealed that enlarged LDs moved towards the sub-apical domains of WT spheroids (Fig. 4a; Supplementary Movies 4, 5),

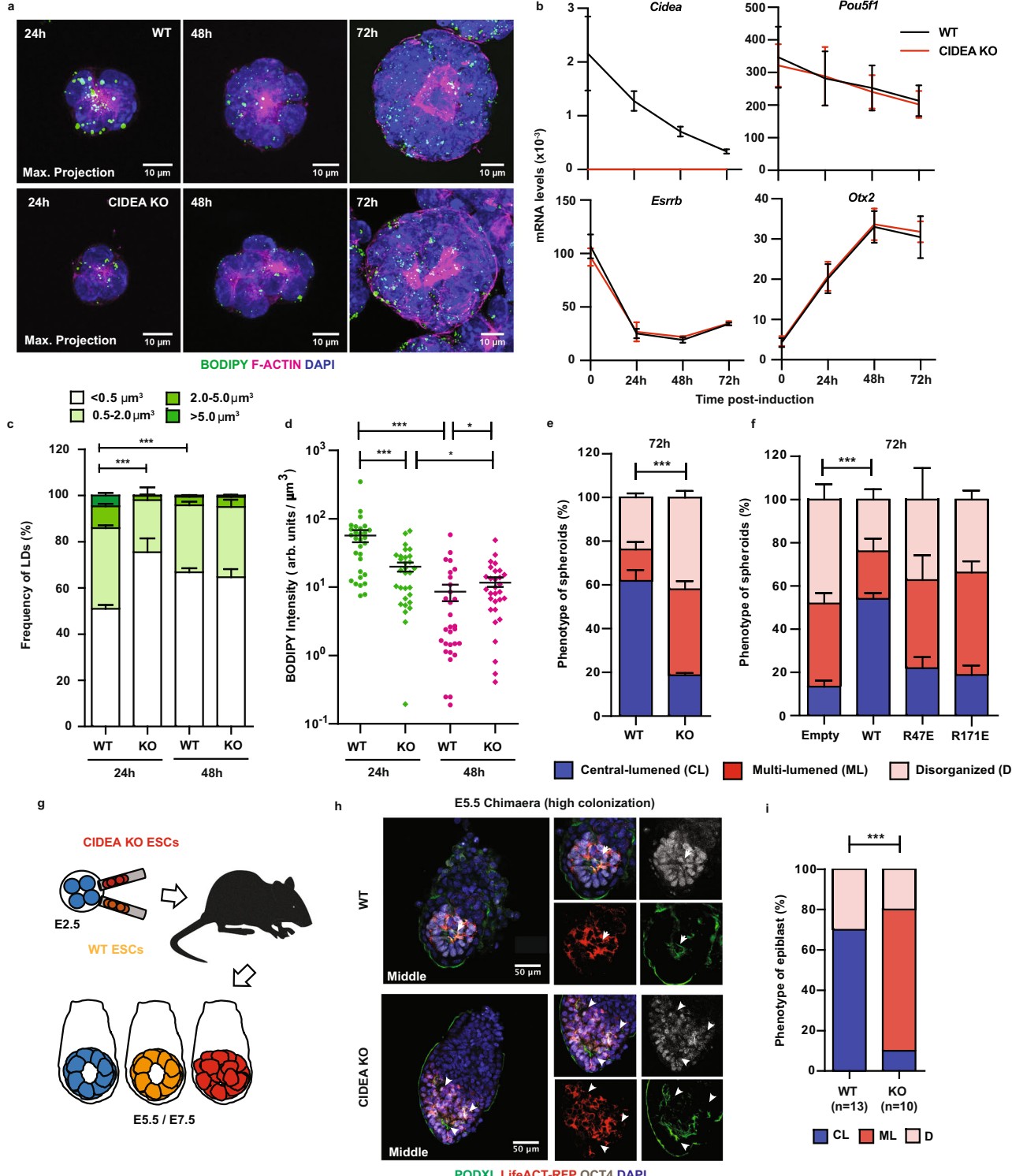

mirroring the repositioning of other organelles (Supplementary Fig. 6d–f)[34,38]. In contrast, BODIPY-labelled LDs did not accumulate apically in polarised KO spheroids, but instead were rapidly lost during the short time-course of our imaging experiment (~2 h; Fig. 4a, right panel; Supplementary Movie 6). High turn-over rate of LDs suggests that promiscuous degradation may operate in the absence of CIDEA-mediated LD enlargement, concurring with the reduced levels of stored lipids we detected in CIDEA KO spheroids (24 h; Fig. 3d and Supplementary Fig. 5a).

To corroborate this view, we treated KO spheroids from induction with the inhibitor of cytosolic lipases diethylumbelliferyl phosphate (DEUP)[39,40] and quantified total BODIPY intensity 24 h later. The treatment rescued, at least partially, the lipid storage capacity of KO spheroids through cell polarisation (see Fig. 4b, c), and this rescue effect was sufficient to improve the efficiency of forming centrally-lumened KO spheroids upon subsequent differentiation (Fig. 4d, e). Together, our results indicate a key role for CIDEA-promoted LD fusion in protecting

**Fig. 3 Abolishing CIDEA-mediated lipid storage impairs morphogenesis. a** Z-stack images of BODIPY 493/503-stained LDs in spheroids formed by WT and CIDEA KO (clone 19.7) ESCs ($n = 3$). Apical domains of spheroids are marked by concentrated F-ACTIN labelled with phalloidin. Similar data were obtained with clones 16.1 and 19.9. **b** *Cidea* transcript expression (RT-qPCR) along with the pluripotency *Pou5f1* (also known as *Oct4*), *Esrrb* (naive), and *Otx2* (primed) factors in WT (black) and CIDEA KO (red) ESCs prior to induction (0 h) and post-induction of spheroid differentiation (24, 48 and 72 h). Error bars, means ± s.e.m. ($n = 3$) (see also Supplementary Fig. 5b–e for additional RNA-sequencing analysis). Quantification of **c** LD volume ($\mu m^3$) and **d** neutral lipid content (BODIPY fluorescence intensity normalised to spheroid volume) in spheroids from (**a**). Each symbol in (**d**) represents one single spheroid at 24 h (green) and 48 h (magenta) post-induction. 30 spheroids per condition and time-point were analysed across independent experiments ($n = 3$). Error bars, means ± s.e.m.; two-sided Mann–Whitney U test; *$p < 0.05$; ***$p < 0.001$. Exact $p$-valued: **c** WT 24 h vs CIDEA KO 24 h, $p = 2.1 \times 10^{-13}$; WT 24 h vs WT 48 h, $p = 1.2 \times 10^{-8}$, and **d** WT 24 h vs CIDEA KO 24 h, $p = 0.0002$; WT 48 h vs CIDEA KO 48 h, $p = 0.0242$; WT 24 h vs WT 48 h, $p = 3.5 \times 10^{-8}$; CIDEA KO 24 h vs CIDEA KO 48 h, $p = 0.0364$. **e** Quantification of the phenotype of WT and CIDEA KO spheroids 72 h post-induction of differentiation, scored as central-lumened (CL), multi-lumened (ML) and disorganised (D) (see also Supplementary Fig. 6a,b). $n \geq 137$ spheroids; error bars, means ± s.e.m. ($n = 3$); two-sided Chi-square (Fisher exact) test; ***$p < 0.001$ ($p = 8.6 \times 10^{-13}$). **f** Quantification of the phenotype of spheroids formed by CIDEA KO ESCs stably transfected with control (Empty), wild-type (WT) and point-mutated (R47E and R171E) V5-tagged Cidea constructs (see also Supplementary Fig. 3b; Supplementary Fig. 6g). $n \geq 177$ spheroids; error bars, means ± s.e.m. ($n = 3$); two-sided Chi-square (Fisher exact) test; ***$p < 0.001$ ($p = 2 \times 10^{-11}$). **g** Experimental set-up. LifeAct RFP-expressing WT and CIDEA KO ESCs were injected in E2.5 embryos and transferred back into surrogate females. Embryo chimaeras were harvested and analysed at E5.5 or E7.5. **h** Immunostaining of representative highly colonized E5.5 ESC (labelled by RFP) embryo chimaeras ($n = 10$–13 embryos). Epiblast is labelled by the lineage marker OCT4. White arrows, foci of lumen formation marked by PODXL (see also Supplementary Movies 2, 3). **i** Quantification of the epiblast phenotype of E5.5 embryo chimaeras from (**h**) scored as central-lumened (CL), multi-lumened (ML) and disorganised (D). Embryo numbers: WT, CL = 9/13, ML = 0/13, D = 4/13; CIDEA KO, CL = 1/10, ML = 7/10, D = 2/10. Two-sided Chi-square (Fisher exact) test; ***$p < 0.001$ ($p = 0.00095$). Source data are provided as a Source data file.

LDs against immediate degradation and reinforce the proposed relationship between safeguarded lipid storage and morphogenesis events.

**The mobilization of stored lipids into lysosomes forms an integral part of lumenogenesis.** Given that LD-stored lipids are ultimately mobilized in the developing epiblast both in vivo and in vitro in a time-dependent manner, we next evaluated the functional importance of the two known mechanisms of LD degradation (i.e., lipolysis and lipophagy) using a pharmacological approach. Lipolysis is mediated by cytosolic lipases recruited at the surface of LDs including the rate-limiting adipose triglyceride lipase (ATGL)[18]. As we here focussed on the mobilization of enlarged LDs, which accumulate normally in polarised WT spheroids, drugs were introduced from 24 h onwards as outlined in Supplementary Fig. 7a. Inhibiting ATGL activity using Atglistatin (ATGLi)[41] had little effect on the mobilization of LDs (48 h; Supplementary Fig. 7b) and formation of central-lumened spheroids (Supplementary Fig. 7c, d). Similarly, the incidence of normally formed lumens remained mostly unaffected upon treatment with DEUP (72 h; Supplementary Fig. 7c, d), excluding a prime role for lipolysis in the regulation of lumenogenesis. Therefore, we turned our attention to lipophagy as an alternative mechanism of lipid mobilization, which involves the autophagic degradation of LDs upon recruitment into lysosomes[42]. In striking contrast to lipolysis inhibition, we found that preventing the lysosomal hydrolysis of LDs upon treatment with the lysosomal acid lipase inhibitor LAListatII (LALi)[43,44] blocked the mobilization of neutral lipids as reflected by high retention of BODIPY signals (Fig. 5a; see also Supplementary Fig. 7b, right panel) and furthermore prompted the formation of multi-lumened or disorganised spheroids 48 h and 72 h post-induction of differentiation (Fig. 5b; Supplementary Fig. 7c, d). These results convincingly point to a critical role for lipophagy-mediated mobilization of LD-stored lipids at the onset of lumen formation with impact on the morphogenesis of pluripotent cells.

In agreement with this conclusion, enlarged LDs (BODIPY; green) were shown to interact with lysosomes (LysoTracker; red) in ESC-induced spheroids (24 h; Fig. 5c) as opposed to mitochondria (via ATPB; see Supplementary Fig. 6e, f for comparison). In peri-implantation embryos, we also observed a significant increase in the co-localisation of LDs with lysosomes in the epiblast of E4.5 relative to E3.5 blastocysts (Fig. 5d, e),

coinciding with the onset of LD mobilization in vivo. This was mirrored by an increased incidence of LD co-localisation with LC3, an autophagosome marker (Supplementary Fig. 8a–c), confirming the involvement of autophagosomes, engulfing LDs, before fusion with lysosomes. For direct evidence of LD autophagic degradation, we next examined the dynamics of LDs and the phenotype of spheroids upon genetic depletion of the autophagy-related protein ATG5[45] and hence inhibition of autophagic fluxes (Supplementary Fig. 8d). WT and ATG5 KO spheroids similarly formed enlarged LDs (24 h; Fig. 5f), validating that the loss of autophagy did not compromise the accumulation of stored lipids upon cell polarisation. ATG5 KO spheroids, however, showed a clear retention of enlarged LDs (48 h; Fig. 5f) and neutral lipid content (48 h; Supplementary Fig. 8e, f), and displayed abnormal morphology upon differentiation (Fig. 5g, h; see also Supplementary Fig. 8g, h), thus recapitulating the effect of LAListatII treatment on WT spheroids.

Collectively, our results unveil how the mobilization of stored lipids via lipophagy supports the proper execution of epiblast lumenogenesis, and furthermore highlight the intricacy of LDs, as central organelles, in the regulation of cell state transitions by bridging metabolism and cell biology.

## Discussion

Across mammalian species, oocytes and pre-implantation embryos have been reported to accumulate LDs[46], which may be physiologically important during this developmental window. Notably, dynamic changes in LD morphology and enlargement were observed upon mouse blastocyst development (refs. [24,25] and this study). This coincides with the activation of the tricarboxylic acid cycle (TCA), which contributes to de novo fatty acid synthesis[47–50]. Besides the endogenous source, early embryos can also take up fatty acids from reproductive tract fluids[22], which are used for β-oxidation to yield energy[51] and for storage as neutral lipids[21,23]. Most recently, a small number of studies attempted to address the functional importance of LDs in pre-implantation embryos via enforced autophagy of the organelles[52] and pharmacological inhibition of lipid synthesis[23], resulting in developmental arrest and/or decreased embryonic viability. While these reports established the importance of maintaining the proper amount of stored lipids for the formation of the blastocyst, the implication of building-up LDs for subsequent development was not investigated.

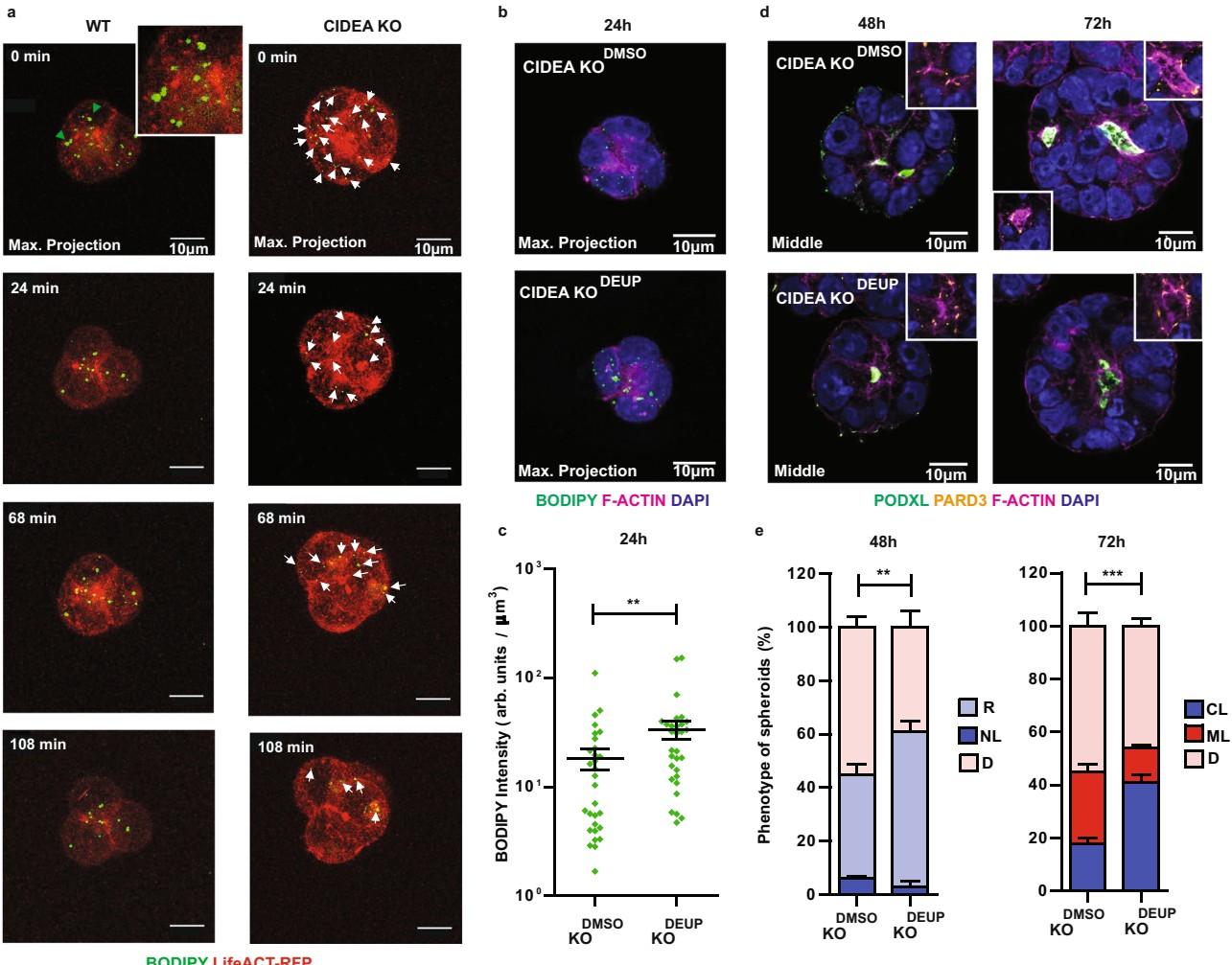

**Fig. 4 CIDEA-mediated LD enlargement confers protection against promiscuous degradation by cytosolic lipases. a** Time-lapse live imaging of BODIPY 493/503 pulse-labelled LDs in spheroids formed by LifeACT-RFP-expressing WT and CIDEA KO ESCs 24 h post-induction of differentiation (see also Supplementary Movies 4–6). White arrows indicate small LDs being degraded in CIDEA KO spheroids. Representative images of ~9 spheroids across independent experiments ($n = 7$). **b** Representative images of BODIPY-stained LDs in CIDEA KO spheroids treated with DMSO (CIDEA KO$^{DMSO}$) and the cytosolic lipase inhibitor DEUP (CIDEA KO$^{DEUP}$) for 24 h upon induction of differentiation ($n = 3$). DEUP, Diethylumbelliferyl phosphate. **c** Quantification of neutral lipid content (BODIPY fluorescence intensity normalised to spheroid volume) in spheroids from (**b**). $n = 30$ spheroids per condition and time-point. Error bars, means ± s.e.m. ($n = 3$); two-sided Mann–Whitney U test, **$p < 0.01$ ($p = 0.0053$). **d** Representative images of CIDEA KO$^{DMSO}$ and CIDEA KO$^{DEUP}$ spheroids at indicated time-points. Foci of lumen formation are marked by PODXL, PARD3 and F-ACTIN ($n = 3$). **e** Quantification of the phenotype of spheroids from (**d**). $n \geq 168$ spheroids; error bars, means ± s.e.m. ($n = 3$); two-sided Chi-square (Fisher exact) test, **$p < 0.01$ (48 h, $p = 0.0016$); ***$p < 0.001$ (72 h, $p = 8 \times 10^{-7}$). R, rosette; NL, nascent lumen; CL, central-lumened; ML, multi-lumened; D, disorganised. Source data are provided as a Source data file.

In our study, we delved into this question by investigating the mechanisms that govern the dynamics of LDs in the epiblast of the implanting blastocyst. Combining embryo examinations with ESC-derived spheroid models, we discovered a previously unrecognised link between the sequential enlargement and mobilization of LDs, and morphogenesis of pluripotent cells. Interruption of this sequence, elicited by premature loss of lipid storage or blocking lipophagy-mediated LD mobilization, induces the aberrant formation of multiple lumens within disorganised epithelial structures (Fig. 5i), demonstrating the functional significance of stored lipids. Interestingly, similar dynamics of lipid storage is observed in the polar TE (Supplementary Fig. 4c, d), which gives rise to the extra-embryonic ectoderm (ExE) and undergoes lumenogenesis[53,54]. While it may be tempting to speculate a conserved role for LD-stored lipids in embryonic and extra-embryonic tissues, the timing and mechanisms of the future ExE morphogenesis differ from that of the epiblast, calling for further investigations using relevant stem cell-based models[54–56] in conjunction with in vivo examinations.

In the pluripotent compartment, we mechanistically identified an essential role for the adipocyte enriched CIDEA protein in regulating LD enlargement to optimise and safeguard lipid storage prior to implantation. We find that abrogating CIDEA's function promotes the cytosolic degradation of LDs by lipases (lipolysis), which in turn reduces the lipid storage capacity of the cells. Importantly, this reduction affects lumenogenesis, as preventing lipid degradation in CIDEA KO spheroids from the onset of differentiation improves the proportion of centrally formed lumens. Although it remains unclear how CIDEA or its LD fusion activity confers protection against promiscuous LD degradation, it has been shown that CIDEC, a closely related protein to CIDEA, interacts with ATGL via its LD clustering and fusion

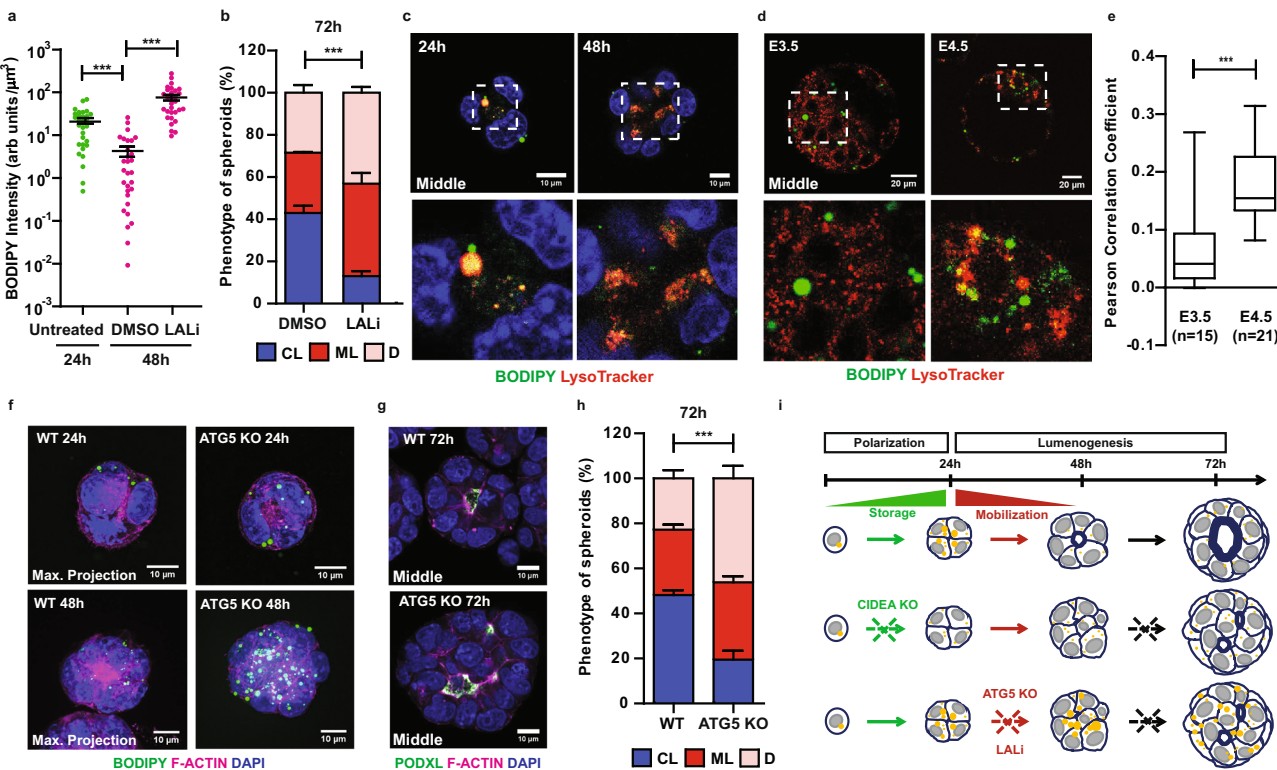

**Fig. 5 Lipophagy-dependent mobilization of lipid storage is required for polarity-driven lumenogenesis. a** Quantification of neutral lipid content using BODIPY 493/503 staining (BODIPY fluorescence intensity normalised to spheroid volume) in WT spheroids before (24 h; green) and after (48 h; magenta) culturing with DMSO or LAListatII (LALi) (see also Supplementary Fig. 7a, b). $n = 30$ spheroids per condition and time-point. Error bars, means ± s.e.m. ($n = 3$); two-sided Mann–Whitney U test; ***$p < 0.001$; exact p-values: untreated 24 h vs DMSO 48 h, $p = 2.4 \times 10^{-6}$; DMSO 48 h vs LALi 48 h, $p = 0.00012$. **b** Quantification of the phenotype of WT spheroids (72 h) after culturing with DMSO or LALi for 48 h (see also Supplementary Fig. 7c, d). $n \geq 185$ spheroids; error bars, means ± s.e.m. ($n = 3$); two-sided Chi-square (Fisher exact) test; ***$p < 0.001$ ($p = 5.6 \times 10^{-14}$). CL, central-lumened; ML, multi-lumened; D, disorganised. **c** Representative live-cell imaging of LysoTracker-labelled lysosomes, BODIPY-labelled LDs, and Hoechst 33342-labelled nuclei in WT ESC-induced spheroids. Squares indicate magnified regions. The analysis was performed on at least two independent experiments ($n \geq 2$). **d** Representative images of lysosomes and LDs in E3.5 (~98 h post-hCG injection) and E4.5 (~122 h post-hCG injection) blastocysts stained as in (**c**). **e** Co-localisation analysis of fluorescence signals from (**d**). Pearson Correlation Coefficient (+1, correlation to −1, anti-correlation); two-sided Mann–Whitney U test; ***$p < 0.001$ ($p = 2.6 \times 10^{-13}$); $n = 15$–21 blastocysts. Box limits = 25th−75th percentiles; centre lines = medians; whiskers end = min. and max. values. **f** Images of BODIPY-stained LDs in spheroids formed by WT and ATG5 KO ESCs at indicated time-points ($n = 3$). **g** Representative images of spheroids formed by WT and ATG5 KO ESCs at 72 h post- induction of differentiation. Foci of lumen formation are marked by PODXL and F-ACTIN. **h** Quantification of the phenotype of WT and ATG5 KO spheroids from (**g**) (see also Supplementary Fig. 8g, h). $n \geq 167$ spheroids; error bars, means ± s.e.m. ($n = 3$); two-sided Chi-square (Fisher exact) test; ***$p < 0.001$ ($p = 1 \times 10^{-8}$). CL, central-lumened; ML, multi-lumened; D, disorganised. **i** Scheme model summarising the findings of this study. Source data are provided as a Source data file.

domain to suppress lipolysis activity in adipocytes[57]. More generally, LD enlargement and related alteration of the LD-surface proteome are thought to limit the recruitment and/or activation of lipases[58]. Therefore, investigating in the future the interacting partners of CIDEA and LD-surface proteome during differentiation may shed light on how lipid storage and its timely mobilization are regulated in pluripotent cells.

In vivo, we validated that the developing post-implantation epiblast also forms multiple lumens in the absence of CIDEA as observed in highly colonised CIDEA KO embryo chimaeras at ~E5.5 ($n = 7/10$). The phenotype was, however, resolved when chimaeras were allowed to reach the early bud stage (~E7.5; $n = 3/3$), indicative of developmental compensation. This agrees with the non-lethality of CIDEA KO mice[35], albeit the full impact of CIDEA depletion on the normal incidence of embryo development and survival (e.g., number of littermates) has not been reported in this model. Interestingly, Madin-Darley Canine Kidney (MDCK) cell aggregates, a well-established epithelial cell model, have been shown to form multiple lumens by apoptosis upon cell polarity disruption. These microlumens subsequently

merge into a single cavity; a sequence of events similarly observed during the formation of the pro-amniotic cavity in increased-size embryos[37]. Hence, a switch from polarity-driven to apoptosis-driven mechanisms may also occur during the remodelling of the epiblast in the absence of CIDEA, as signified by an increased incidence of apoptotic events in CIDEA-deficient spheroids although this has not been formally quantified in this study.

Another key finding of our study is the importance of lipophagy in the process of polarity-driven lumenogenesis. Pharmacological inhibition of lysosomal acidic lipases along with our complementary analysis of autophagy deficient (ATG5 KO) models established that lipophagy (as opposed to lipolysis) is strictly required for the timely mobilization of stored lipids into lysosomes and the proper formation of an apical lumen. Importantly, this result not only implies that the significance of lipid storage lies on its subsequent usage, but also demonstrates the specificity of the process. While ATG5 depletion inhibits autophagic fluxes and LD mobilization, the initial building-up of stored lipids is not impacted. This highlights that the two phases of lipid 'management' (i.e., storage and mobilization) are independently

regulated yet are required together for proper lumenogenesis (Fig. 5i). Similarly, we confirmed that autophagy fluxes remain active in the absence of CIDEA (see Supplementary Fig. 8d), although we cannot exclude that the altered repositioning of organelles in KO spheroids may, at least partly, hamper lipophagy via reduced LD-lysosome interactions, as also suggested by the sporadic retention of high neutral lipid content in differentiating CIDEA-deficient spheroids (Supplementary Fig. 5a).

At the present, we still know relatively little of the signalling pathways that control lipid storage and its degradation coinciding with the hollowing of a central lumen. One potentially significant candidate is the AMP-activated protein kinase (AMPK), a well-established energy sensor and key modulator of lipid metabolism[59–61]. Whether AMPK activity is transiently modulated in the implanting blastocyst remains, however, to be studied. Of relevance, blastocysts temporarily arrested in diapause (i.e., embryos awaiting uterine receptivity to implant) have been reported to shift their metabolism from reliance on carbohydrates to LD-derived fatty acids[46]; a process found to be associated with enhanced lipolysis upon sustained activation of AMPK and inactivation of mTORC2[62]. In this context, lipolysis-mediated LD degradation was proposed to fuel β-oxidation to support the survival of developmentally paused embryos. In contrast, we find that inhibiting lipolysis (Supplementary Fig. 7) or β-oxidation (Mau, KHT & Azuara, V, unpublished observations) does not impede the morphogenesis of 3D spheroids as opposed to LAListatII treatment, highlighting different regulatory mechanisms and functions for LDs at peri-implantation times. While lysosomes are known to actively degrade and recycle intracellular materials including LDs, they also emerged as central hubs for intrinsic signalling pathways driven by metabolite availability[63]. Hence, determining whether critical LD-derived lipids are retained within lysosomes as metabolic signals and/or redistributed into different cellular compartments will be crucial to unfold how downstream events of lipophagy may coordinate developmental progression and morphogenesis.

In summary, our findings unveil how lipid storage trafficking and morphogenesis of pluripotent cells are fundamentally connected in peri-implantation embryos. Given the importance of lipid metabolism and cellular remodelling in development and tissue homeostasis, we believe that studying these two processes as being closely interrelated will be of relevance for both basic and clinical research.

## Methods

**Mouse embryos.** Laboratory animals used in this study are of the species Mus musculus (CD1 and C57BL/6J strains). Mice were housed in a 12-h dark and 12-h light cycle. CD1 females were crossed with either C57BL/6J or CD1 males to obtain stage-specific embryos. Embryos were dissected in M2 medium (Sigma-Aldrich, M7167). For natural mating, sexually mature CD1 females were used (usually 6-week to 3-month-old). For superovulation, 5 international units (iu) Pregnant Mare Serum Gonadotrophin solution (Folligon, MSD Animal Health) was administered to 3–4-week-old females via intraperitoneal injection. After 2 days, 5 iu human Chorionic Gonadotrophin solution (hCG; Chorulon, MSD Animal Health) was injected, followed by crossing with males (usually 3–6-month up to 18-month-old). Six females were used per experiment. All experiments performed in the U.K. were approved by the Home Office (project licences to M.P. and S.S.) and have been regulated by the Animals (Scientific Procedures) Act 1986 Amendment Regulations 2012 following ethical reviews by the Animal Welfare and Ethical Review Body (AWERB) at Imperial London College and the Local Ethical Review Panel (LERP) at the Department of Physiology, Anatomy and Genetics of the University of Oxford. Animal procedures performed in France were carried out according to French national rules on Ethics and Animal Welfare in the Animal Facility (IERP, INRAE, 10.15454/1.5572427140471238E12, Jouy-en-Josas). This work was approved by the French Ministry of Higher Education, Research, and Innovation (n°15–78) and the local Ethical Committee (INRAE Jouy-en-Josas Centre).

**Pre-implantation in vitro culture (IVC).** Four to eight cell embryos were collected at ~68 h post-hCG injection and cultured to E2.5 morula for 6 h (~74 h post-hCG

injection), or E3.5 blastocysts for another 24 h (~98 h post-hCG injection). Alternatively, E3.5 blastocysts were collected at ~98 h post-hCG injection and cultured to E4.5 for 24 h (~122 h post-hCG injection). All embryos were cultured in EmbryoMax KSOMaa medium (Sigma-Aldrich, MR-106) under mineral oil, at 37 °C under 5% $CO_2$. Embryos were then fixed by 4% paraformaldehyde (PFA) and proceeded with immunostaining and LD staining.

**Mouse ESC-embryo chimaeras.** Host embryos were collected after natural mating at 2-cell stage and cultured overnight in M16 medium (Sigma-Aldrich, M7292). Embryos of 4–8 cell stages were used for injection the day after. For this, the embryos were transferred to M2 medium under mineral oil on the stage of an inverted microscope, equipped with Narishige micromanipulators. ESCs, initially cultured in 2i/LIF conditions, were trypsinized and re-suspended in M2 medium. A small quantity of cells was placed next to the embryos. Six to ten cells were injected into the embryos under the zone pellucida by a micro-injector under a piezo impact-driven system. Injected embryos were then transferred to E0.5 pseudo-pregnant CD1 female mice. Embryo chimaeras were harvested from foster females at E5.5 or E7.5 equivalent times.

**Cell culture.** All cell cultures were maintained at 37 °C under 5% $CO_2$. ESCs were routinely cultured on 0.1% gelatine coated culture plates in GMEM-BHK12 basal medium (Gibco, 21710-025) supplemented with 10% foetal bovine serum (FBS) (Gibco, 10270-106), Leukaemia Inhibitory Factor (LIF) (made in-house), 0.1 mM ß-mercaptoethanol (Gibco, 31350010), 0.24% (w/v), sodium bicarbonate (Gibco, 25080094), 1 mM sodium pyruvate (Gibco, 11360070), 0.1 mM non-essential amino acids (Gibco, 11140050), 2 mM L-glutamine (Gibco, 25030149) and 25 U/ml Pen-strep (Gibco, 11548876). Alternatively, ESCs were cultured on 0.2% gelatine coated culture plates in DMEM F12: Neurobasal (vol. ratio 1:1) medium (Gibco, 11320033 and 21103049) supplemented with N2 (Gibco, 17502048), B27 (Gibco, 17504044), 0.1 mM ß-mercaptoethanol, 2 mM L-glutamine and 25 U/ml Pen-strep, with inhibitors CHIR99021 (3 µM) (Tocris, 4423), PD0325901 (1 µM) (Tocris, 4192) and LIF, known as 2i/LIF conditions. For KSR (ThermoFisher) or AlbuMAX (ThermoFisher, 11020021) treatment, cells were seeded the day before, followed by replenishing medium with FBS replaced by 15% KSR or specific concentration of AlbuMAX, and cultured overnight.

**Self-renewal and proliferation assays.** For assaying the self-renewal capacity of ESCs, alkaline phosphatase (AP) activity was tested. In brief, cells were seeded at 1000 cells/ml in each well and grown for 7–8 days, with medium replenished every day. AP activity was assessed using the AP Staining Kit (Sigma-Aldrich, SCR004) according to the manufacturer's instructions. Stained colonies were counted and categorized into undifferentiated, mixed and differentiated based on morphology and staining intensity. For testing the proliferation of ESC cultures, cells were seeded at $3–5 × 10^5$ cells per well. After 2 days, cells were dissociated, counted and $3–5 × 10^5$ cells were re-plated. This was repeated for 8 passages and the growth curve was determined.

**ESC spheroid formation.** ESC spheroids were formed as fully described in ref. [1], with minor modifications. Briefly, ESCs were dissociated by TrypLE (Gibco, 12604013) and counted on a haemocytometer. Cells were washed with DPBS (Sigma-Aldrich, D8537) and re-suspended in ice-cold Growth Factor-reduced Matrigel (Corning, 354230) as a single-cell suspension at 1500 cells per µl. The Matrigel cell suspension was seeded as 25 µl drops and set in ibidi-µ plate 8-well chambers (ibidi, 80821) at 37 °C for 10 min before replenishing FBS (15%) supplemented medium without LIF (-LIF). For experiments with pharmacological inhibition, sources and concentration of the inhibitors can be found in Supplementary Information (see Supplementary Data 1). For molecular experiments, cells were seeded as a "3D-on-top" culture. Briefly, pre-chilled culture plates were coated with Matrigel and set at 37 °C for 30 min. Single-cell suspension in culture medium (15% FBS; -LIF) was then seeded at $5 × 10^4$ cells per $cm^2$. Once the cells are attached, the medium was replaced by 5% Matrigel supplemented medium (5% Matrigel; 15% FBS; -LIF). For collecting the spheroids, Matrigel cultures were washed with DPBS, followed by gentle scraping and incubation in chilled Cell Recovery Solution (Corning, 354253) for 30 min. The suspension was then collected in equivalent volume of cold PBS, followed by centrifugation ($150 × g$) to pellet the cells. Pelleted spheroids were washed in cold PBS again, snap-frozen before being processed for subsequent analyses.

**Molecular cloning.** CIDEA WT and point-mutation (R47E and R171E) constructs were previously generated as detailed in Barneda et al.[27]. Insertion fragments were amplified by PCR (list of primers can be found in Supplementary Data 1), purified by PCR purification kit (Roche, 11732676001), followed by restriction enzyme digestion. Digested fragments with sticky ends were purified by gel electrophoresis and extracted by gel extraction kit (NEB, T1020S). Purified inserts were ligated to pre-digested pPyCAGIP plasmids by DNA ligase (NEB, M020S) at 16 °C overnight. Ligation product was transformed into DHFalpha competent bacterial cells and amplified. Bacteria culture was lysed, and plasmids purified by midiprep plasmid extraction kits (Invitrogen, K210004), followed by Sanger sequencing for verification.

**CRISPR-Cas9**. To generate CIDEA KO ESCs, two sgRNAs were designed by the MIT online tool (http://crispr.mit.edu), to target the first intron and the last exon, leading to the complete deletion of the *Cidea* locus. Oligos encoding the sgRNAs were inserted into px458 and px459 plasmids, which also encode the Cas9 protein (Addgene, gifted by Feng Zhang). Plasmids were transfected into ESCs by nucleofection (Lonza, VPH-1001) following the manufacturer's protocol. Successfully transfected cells were selected by transient antibiotic resistance and fluorescence expression. These cells were then sub-cloned manually and PCR (NEB, M0482; 30 cycles of 94 °C for 30 s, 60 °C for 30 s and 68 °C for 40 s) genotyped using primer pairs flanking the edited loci (see Supplementary Data 1). A scheme of gene-editing strategy and genotyping can be found in Supplementary Fig. 2d.

**Transfection**. Cells were seeded at $2 \times 10^5$ in a well of 6-well plate, the day prior to the transfection. On the day of transfection, the reaction mix was prepared by mixing 100 μl OptiMEM (Gibco, 31985026), 1 μg DNA and 1 μl Lipofectamine 2000 (ThermoFisher, 11668027). The mix was left to form a complex at room temperature for 10 min. Medium was replaced with 1.5 ml fresh media with addition of 100 μl of the transfection mix. The day after, cells were washed and replenished with fresh medium in the presence of puromycin for selecting successfully transfected cells. For generation of sub-clones, cells were seeded at low density and allowed to grow into individual colonies. Colonies were manually isolated and expanded as independent cell lines.

**Immunostaining**. Cells were fixed in either 2% (embryo chimaeras) or 4% PFA for 10–30 min (10 min for cell culture, 20 min for ESC spheroids and 15–30 min for embryos). Fixed samples were washed with PBS three times, 5 min each, followed by permeabilization by 0.3–0.5% Triton-X or 0.05% Saponin in 10% FBS-PBS, for 10–30 min. Permeabilized cells were washed 3 times, followed by incubation with primary antibody in 10% FBS-PBS overnight at 4 °C (for a complete list of antibodies and dilution factors, please see Supplementary Data 1). The day after, samples were washed by PBS, followed by incubation with the secondary antibodies (Alexa Fluor conjugated secondary antibodies) and fluorescently labelled phalloidin (F-ACTIN staining) for 1 h at room temperature. Samples were then washed and incubated with 40 μg/ml DAPI (Sigma-Aldrich, D9542) for nuclei staining. Samples were mounted in Prolong (Invitrogen, P36930/P36970) or Vectashield mounting agent (Vectorlabs, H1000). Most experiments were imaged using a Leica SP5 confocal microscope. Peri-implantation embryos for lineage-specific LD analysis were imaged by Zeiss LSM710 confocal microscope. Embryo chimaeras were imaged with an inverted ZEISS AxioObserver Z1 microscope equipped with an ApoTome slider, a Colibri light source, Axiocam MRm camera and driven by the Axiovision software 4.8.2. The full z-stack sections of WT and CIDEA KO embryo chimaeras shown in Fig. 3h can be found in Supplementary Movie 2 and Movie 3, respectively. List of all antibodies used with company names, catalogue numbers and dilution factors can be found in Supplementary Data 1.

**LD staining**. Fixed samples were stained with BODIPY 493/503 (5 μg/ml in PBS protected from light exposure) (Invitrogen, D3922) for 30 min. Samples were then washed 3 times with PBS, followed by DAPI staining for 10 min. BODIPY staining was carried out after secondary antibody incubation, in the case of staining LDs in immunostaining samples.

**Lipophagy live-cell imaging**. Embryos or ESC spheroids were co-stained for lysosomes and LDs with LysoTracker Red (500 nM) (Invitrogen, L7528) and BODIPY 493/503 (5 μg/ml), respectively. Cells were first washed and stained for lysosomes by incubation in medium with the addition of LysoTracker for 15 min. After lysosome labelling, cells were washed three times with PBS, followed by LD labelling in PBS with BODIPY 493/503 for 10 min. Labelled cells were then washed with PBS and replenished with culture medium (in the presence of Hoechst33342, 1:1000 for visualizing nuclei; Thermo Scientific, 62249). Labelled embryos or spheroids were then imaged on a Leica SP5 confocal microscope in ibidi μ-plates.

**Time-lapse microscopy**. For time-lapse imaging, cells were seeded and imaged in phenol-red free culture medium supplemented with 50 mM HEPES. Imaging was carried out by a Leica SP5 confocal microscope in imaging cabinet maintained at 5% $CO_2$ and 37 °C. The imaging stage was set up for multiple-position capturing and z-stacks were captured at 1 μm per step. For imaging LDs, live cells were pulse-labelled by BODIPY 493/503 for 10 min, followed by washes and replenished with pre-warmed culture medium.

**Image analysis**. LD size and number in cultured pre-implantation embryos were measured by Fiji ImageJ using the "3D object counter" plug-in to analyse the BODIPY channel. Image stacks were first processed by "3D fast filter" to minimize background signal prior to analysis. LD size and number of ESC spheroids were analysed using the same method. To measure the neutral lipid content (i.e., total LD volumes) of ESC spheroids using BODIPY staining, the fluorescence intensity of each individual LD was measured using the same "3D object counter" plug-in. Neutral lipid content was then calculated as the sum of the fluorescence intensity of all LDs in each spheroid and standardized to the spheroid volume. Volume of the spheroids was measured by the same plug-in on F-ACTIN channel, which marks the cell boundary of the spheroids. Lineage-specific analysis of LD size and number in freshly harvested peri-implantation embryos was carried out using the Imaris v6.3 software (Bitplane). Briefly, embryonic tissues were manually outlined on each image plane of the z-stack images, according to E-CADHERIN staining. The 3D structures of the tissue were then re-constituted by the software and LDs in each tissue structure were selected and analysed under automatic segmentation. Details of the lineage-specific LD analysis pipeline can be found in Supplementary Fig. 4a, b. For LD analysis in ESC cultures, Cell Profiler software was used and the "identify primary objects" plug-in was used to determine the number and area of each LD. Co-localization of fluorescence signals, including LysoTracker/BODIPY (see Fig. 5d, e) and LC3/BODIPY (Supplementary Fig. 8b, c), was assessed by "Ez Colocalization" plug-in[64] with Fiji ImageJ, and presented as Pearson Correlation Coefficient.

**Triglyceride (TG) quantification**. ESCs or spheroid cultures were collected as described above (see the 'Cell culture' and 'ESC spheroid formation' sections, respectively). Samples were washed in PBS and snap-frozen in liquid nitrogen before processing. Before the assay, cell number equivalents of snap-frozen cell lysate samples were determined by CyQuant Cell Proliferation Kit (Invitrogen, C7026) according to the manufacturer's manual, and $1 \times 10^6$ cell equivalents were taken for the assay. Triglyceride levels of cell lysate samples were measured using the Triglyceride Colorimetric Assay Kit (Cayman, 10010303) on a fluorescent plate reader according to the manufacturer's instruction.

**RNA extraction**. Cells were collected and lysed in RLT buffer. RNA extraction was carried out by RNeasy Mini Kit (Qiagen, 74106) according to the manufacturer's instruction. RNA eluted from the extraction columns were analysed by Nanodrop to determine the purity and concentration.

**RT-qPCR**. Reverse transcription (RT) of RNA was carried out separately, prior to qPCR. For RT, 1 μg of RNA was mixed with 0.5 μl of oligo-dT, 0.5 μl random hexamers and 1 μl of dNTP, as a 11 μl mix. The reaction mix was incubated at 65 °C for 5 min and cooled down on ice for 1 min, before adding the SuperScriptIII reverse transcriptase (1 μl; Invitrogen, 18080044) with 1 μl of 0.1 M DTT, 4 μl of 5x reaction buffer and 1 μl of RNaseOUT (Invitrogen, 10777019). The mixture was incubated at 25 °C for 5 min followed by 50 °C incubation for 1 h. Reverse transcriptase was then heat-inactivated at 70 °C for 15 min. Resultant cDNA was diluted to 5 ng/μl for qPCR. For qPCR, 10 ng of cDNA was used in each reaction with 5 μl of SYBR Green qPCR reagent (Merck, kcqs00) and 0.01 nmol/μl primers. qPCR was run as followed, 95 °C for 15 min; 40 cycles of 94 °C for 15 s, 60 °C for 30 s and 72 °C for 30 s. For analysis, Ct value of analysed genes was normalized to the average of Ct values of the two housekeeping genes L19 and S17. List of primers used can be found in Supplementary Data 1.

**Western blot**. ESCs or spheroid cultures were collected as described above (see the 'Cell culture' and 'ESC spheroid formation' sections, respectively) and lysed in RIPA buffer for protein extraction. Protein concentration was determined by BCA assay (ThermoFisher, 23225) according to the manufacturer's instruction. 10 μg protein was loaded with 4x Lammeli buffer (Bio-Rad, 161-0747) into SDS-PAGE gels (7% running gel/4% stacking gel) with pre-stained protein ladder. After SDS-PAGE, proteins were transferred to the PVDF Immobilon-FL membrane (Millipore, IPL00010) by semi-dry transfer protocol, followed by blocking in 5% milk for 1 h. Blocked membrane was incubated in primary antibody solution (in 5% milk) overnight. The membrane was washed in Tris-buffered saline with 0.1% Tween-20 (TBST), followed by Horseradish peroxidase (HRP)-conjugated secondary antibody (Santa Cruz, sc-516102; dilution of 1:5000) incubation for 1 h at room temperature. The membrane was then washed (1x TBST), developed with Immobilon Forte HRP substrate (Millipore WBLUF0500), and imaged by ImageQuant LAS4000 imager or exposed to X-ray film. Quantification was done by Fiji ImageJ. List of all antibodies and dilution factors used can be found in Supplementary Data 1. Scans of uncropped blots are provided in the Source data file.

**RNA-sequencing and data analysis**. The library preparation and sequencing were conducted by Genewiz. Libraries were sequenced on Illumina HiSeq (2 × 150 bp) to a depth of 30 million reads. Quality filtering of paired-end reads was carried out using FastQC (Simon Andrews, https://www.bioinformatics.babraham.ac.uk/projects/fastqc/), followed by data trimming with Trimmomatic-0.33[65] to remove sequencing adaptors and low-quality bases (<15). Quantification of transcripts was performed using Kallisto v0.43.1[66]. All downstream analysis was carried out in R using Bioconductor packages.

**Statistics and reproducibility**. Statistical analyses were performed using Graph-Pad Prism except RNA sequencing data. No statistical method was used to predetermine sample size of embryos and spheroid analysed and experiments were not conducted under blinded condition. Data were presented as means ± s.e.m. For

qualitative analysis, for example, assessment of spheroid phenotypes, results were statistically tested by two-sided Chi-square (Fisher exact) test, with data plotted as contingency tables. For quantitative analysis, for example, neutral lipid content using BODIPY fluorescence intensity and LD sizes, results were first tested by D'Agostino and Pearson's omnibus normality test, to check for normal distribution. Normally distributed data were then analysed by two-sided unpaired $t$-test with Welch's correction. Non-normally distributed data were analysed by two-sided Mann–Whitney U test. All experiments including quantification experiments were performed in triplicate unless specified. All statistical tests were conducted on data pooled from all replicas of experiments. In cases of data shown as proportions, for example, LD size distribution and spheroid phenotype, results were presented as the mean ± s.e.m. of triplicate experiments. Inter-experiment variation is checked by the same statistical test to ensure consistency of the results.

**Data representation**. Box plot: centre line shows the medians; box limits indicate the 25th and 75th percentiles as determined by GraphPad Prism software; the whiskers end at minimum and maximum values.

**Reporting summary**. Further information on research design is available in the Nature Research Reporting Summary linked to this article.

## Data availability

The RNA-sequencing data generated in this study have been deposited in the Gene Expression Omnibus database under accession code GSE165563. The RNA-sequencing data of mouse embryo development used in this study are available in the ArrayExpress database under accession code E-MTAB-2958. Source data include RT-qPCR data (Fig. 2c; Fig. 3b; Supplementary Fig. 1c, d, f, g; Supplementary Fig. 2a, f, g, h), quantification of immunofluorescence data (Fig. 1b, c; Fig. 2e, f, j; Fig. 3c, d; Fig. 4c; Fig. 5a, e; Supplementary Fig. 4d; Supplementary Fig. 8c, e), records of spheroid phenotypes (Fig. 3e, f; Fig. 4e; Fig. 5b, h; Supplementary Fig. 6b; Supplementary Fig. 7d; Supplementary Fig. 8h), triglyceride assay (Supplementary Fig. 5a; Supplementary Fig. 7f), growth and self-renewal assays (Supplementary Fig. 2i, j, k, l) and uncropped western blots (Supplementary Fig. 2b; Supplementary Fig. 7d). Time-lapse movies described in this study are available as Supplementary Movie 1 (Fig. 2b) and Supplementary Movies 4–6 (Fig. 4a). Z-stack images of representative highly colonized WT and CIDEA KO E5.5 embryo chimaeras are also provided as Supplementary Movies 2, 3 (Fig. 3h). Cell lines and/or plasmids generated in this study are available upon request. Source data are provided with this paper.

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

## Acknowledgements
We thank Noboru Mizushima, Austin Smith, Niall Dillon and Ian Chambers for providing ESC lines constitutively knock-out for *Atg5* and matching controls, ESC-E14TG2a and ESC-R1 lines and the pPyCAGIP plasmid, respectively. Thanks to Megha Prakash-Bangalore, Monica Roman-Trufero, Luc Jouneau and Olivier Jouneau for their technical, statistical and/or graphic design assistance, and to James Elliot at the Imperial/BRC Flow cytometry Facility. Gratitude also goes to Alex Gould, Aylin Hanyaloglu, Tristan Rodriguez, and Victor Zammit for discussions and/or critical reading of the manuscript, and to all members of the Epigenetics and Development group. This work was supported by Genesis Research Trust – U.K. to V.A. (P65250; P54998); Biotechnology and Biological Sciences Research Council – U.K. to V.A. (BB/P005179/1; BB/D52657X/1) and to M.C. (BB/P005209/2; BB/H020233/1); Imperial College President PhD scholarships – U.K. (D.K. and R.A.d.S.); Medical Research Council and UKRI Future Leaders Fellowship – U.K. to Mi.P. (MRC, MC_UP_1605/4 and MC_EX_MR/S015930/1); INRAE and ANR Programme Investissements d'Avenir REVIVE - France (V.B. and A.J., ANR-10-LABX73); Wellcome Senior Investigator Award – U.K. to S.S. (103788/Z/14/Z), Agro-ParisTech and PHASE INRAE - France (Me.P.), University of Oxford – U.K. (S.S.), University of Warwick and Nottingham Trent University – U.K. (M.C.) and Imperial College London – U.K. (V.A. and M.C.).

## Author contributions
K.H.T.M. carried out most experiments and data analyses; D.K., D.B. and Mi.P. performed the functional analysis of CIDEA and its fusion activity in ESCs under the expert guidance of M.C.; D.K and C.R. carried out lineage-specific analysis of LD attributes in peri-implantation embryos under the supervision of S.S.; Mi.P. supervised the analysis of cultured pre-implantation embryos performed by B.L. and K.H.T.M.; V.B. performed microinjection for generating embryo chimaeras; A.J. supervised the generation and analysis of embryo chimaeras and helped experimentally together with Me.P. and K.H.T.M.; R.A.d.S. performed all bioinformatics analyses; V.A. conceived the project and supervised the study; V.A. and K.H.T.M. wrote the manuscript with contributions from all the authors.

## Competing interests
The authors declare no competing interests.
