## [Peer Review File · Nature Communications]

Dynamic Enlargement and Mobilization of Lipid Droplets in Pluripotent Cells Coordinate Morphogenesis during Mouse Peri-implantation DevelopmentREVIEWER COMMENTS

Reviewer #1 (Remarks to the Author):

This manuscript from the Azura lab explores the role of the lipid droplets previously observed in preimplantation and diapause embryos, which are subsequently dispersed after implantation. The authors show that LD size increases during preimplantation development, accompanied by decrease in number, implying LD fusion. They propose that this process is a prerequisite for the morphogenic change in epiblast from apolar to epithelial. CIDEA protein was found to be much more abundant in ESCs compared with MEFs in a screen for likely candidates. OE of Cidea in ESCs caused increase in LD size and decrease in number, whereas KO reduced size and increased number, consistent with its role in LD fusion. The authors further show, using ESCs, that multiple lumens form in 3D aggregates of Cidea KO cells, whereas only one central aggregate formed in WT or Cidea-WT rescue aggregates. WT versus KO ESCs were injected into host embryos and dissected at E5.5. Contribution of KO cells over a threshold level caused disorganisation of the epithelium. Treatment of KO spheroids with DEUP (cytosolic lipase inhibitor) could rescue LDs and (partially) central lumen formation. By testing appropriate inhibitors, the mode of operation for LD lipid release in lumen formation was attributed to lipophagy. Chimaeras generated from in Cidea KO ESCs show multiple lumen formation at E5.5, consistent with the 3D in vitro Cidea KO ESC cultures. However, by E7.5 only one lumen is observed, but whether the embryo is normal is not clear from the poorly presented images in Supp. Fig. 6g. LDs are observed in the VE, which the authors propose may contribute to the rescue of the Cidea KO chimaeras. There must be some other compensatory mechanisms, since Cidea KO in mice does not result in an embryonic lethal phenotype. The methods are very well documented and the statistics seem appropriate. This study is quite interesting but a few improvements need to be made to the manuscript:

1. Fig. 2h: the E3.5 image chosen may not be the best example, since only one of the LDs shows CIDEA protein clearly. Also, it is not clear from the image to which cell this LD belongs. It is very close to the CDX2-positive nucleus, therefore more likely to be TE than ICM. Inclusion of a membrane marker, such as the E-CAD used in Supp. Fig. 4c, would address this question.
2. The qRT-PCRs for pluripotency and differentiation factors presented in Supp. Fig. 2 should be brought into the main figures, since they provide important evidence that the disruption to luminogenesis caused by Cidea KO does not impede exit from naïve pluripotency.
3. In Supp. Fig. 4c the mural TE contains multiple large LDs, but there are apparently none in the polar TE. Could the authors speculate on the destination of these, since it is the polar TE that is required for differentiation of the extraembryonic ectoderm and subsequently the chorion?
4. The chimaera images in Supp. Fig. 6g should be replaced with images in a more conventional orientation with IHC for a couple of lineage markers to aid navigation for the reader.

Reviewer #2 (Remarks to the Author):

In this manuscript, Hang T et al. revealed the dynamic changes of lipid droplets (LDs) and their importance during post-implantation embryogenesis. The authors found that LDs expand after the morula stage due to CIDEA protein-mediated LD fusion, and that lipophagy is involved in the degradation of the enlarged LDs. Using an original spheroid model, they also showed that dysfunction of CIDEA protein and suppression of lipophagy affect lumenogenesis, demonstrating that lipid storage plays an essential role in peri-implantation embryonic development. In general, the evidence in this study is sufficient to support the authors' conclusions, and these results are potentially interesting. However, there are several aspects that need to be resolved/improved before this manuscript can be considered for publication in Nature Communications.

Major points:

1. One of my concerns is that most of the results rely on fluorescent microscopy imaging of LDs. For example, when LD synthesis or degradation is inhibited or autophagic activity is blocked, the author should examine not only changes in LD distribution by BODIPY staining, but also changes in intracellular LD content. For this, Western blotting with antibodies against LD-specific proteins such as Plin2 protein and measurement of intracellular triglyceride content may be effective. This analysis should also be done in experiments with ATG5- and CIDEA-deficient/mutant spheroids.
2. Another point of my concern is the localization of CIDEA protein to the LD surface. In general, it is known that CIDEA protein are enriched at the contact site between LDs to facilitate LD enlargement (as shown in Fig. S3C). However, since the localization of CIDEA protein shown in current manuscript is mostly observed uniformly on the LD surface, the authors need to confirm whether CIDEA protein is enriched at the LD contact sites under the conditions and/or stages when LDs fuse and grow larger.
3. I agree with the authors' view that autophagy (lipophagy) is involved in the degradation of enlarged LDs because the suppression or loss of autophagy results in the accumulation of enlarged LDs. However, the current results are not sufficient to clarify the lipophagy-mediated degradation of the enlarged LDs. The authors claimed the involvement of lipophagy based on the finding that lysosomes are in close proximity to enlarged LDs, but it is also possible that lysosomes directly fuse with LDs. Therefore, in order to further clarify the involvement of lipophagy, it is necessary to 1) observe the localization of autophagosomes in enlarged LDs, and 2) investigate the changes in LD content under conditions of autophagy suppression or deficiency (the second point is related to comment 1).

4. In Fig.S6d, the authors should not only examine the localization of LDs to the Golgi apparatus in the spheroid once expanded LDs are observed, but also to mitochondria and lysosomes if LD mobilization is active at that time.

Minor points:

5. The authors mention "release" of lipids in the title, abstract, and text, but their study does not experimentally demonstrate uptake or release of lipids into LDs, so please reconsider the title and use of "release" in this regard. LD enlargement (via CIDEA protein) and their degradation (by lipophagy) is an important finding, so why not include this in the title?

6. Throughout the manuscript: the developmental time of the fertilized embryos up to the blastocyst stage should not only be indicated by embryonic day, but should also be combined with the time after hCG, for example, to better define the developmental stage. This is because the morphology of LDs seems to change rapidly before and after compaction. Related to this, the exact time (stage) of fertilized embryos indicated as "pre-compaction" should be specified (Figure 1a). Is it 4-cell stage?

7. Currently, it is unclear whether the amount of intracellular LD content actually increases ("storage") or decreases ("mobilization") during spheroid formation, simply based on the morphological changes of LDs by BODIPY staining (as pointed out in comment 1). Depending on the outcome of comment 1, the overview figure in Figs. 5i and S7a should be revised (if necessary) and a more detailed explanation should be provided in the text.

8. P4, line 79: Since the last paragraph contains the results of CIDEA (lines 100-108), please reconsider the subtitle? Alternatively, the paragraph that mentions about CIDEA data could be moved to the next section.

9. Some of the enlarged images (Figs. 2g and 2h, for example) are almost the same size as the original images, so they should be changed to more enlarged image. Along the same lines, CIDEA protein in blastocysts appear to be partially accumulated rather than localized in a "ring" (page 6, line 129) on the LD surface (Fig. 2h). It might be clearer if higher resolution images could be taken. In addition, in some figures (Figs. 1a, 2a, 2d, 2g, 3h, and 5d), the area showing the enlargement and the area of the actual enlarged image are slightly misaligned.

10. Typos: Page 14, line 335: "Narashige" should be Narishige.

Reviewer #3 (Remarks to the Author):

I found this article very interesting. The authors used a range of methods for the study of CIDEA protein in LDs utilization by embryos. It is worthy to highlight that the authors verified their hypothesis on multiple models, analyzing the lipid droplet size formation in the blastocysts, stem cells derived spheroids and finally in the chimeric embryos. To study the role of CIDEA protein, they generated the CIDEA knockout stem cell line (created with the use of CRISPR-Cas 9 methodology) that served for both to study the CIDEA function directly in the embryonic stem cells and during the formation of spheroids and chimeras. Moreover authors used the fluorescence and immunofluorescence to study the CIDEA and lipid colocalization and the real time PCR to study the expression of the key genes involved in the lipid dynamics control.

With the use of specific chemical inhibitors authors confirmed the role of the lipophagy process in the embryonic LD utilization (as opposed to lipolysis that seems to be irrelevant in the embryonic lipid droplets utilization). The finding that CIDEA protein is a potentially key regulator of the LD fusion in embryonic stem cells and in the same time regulates the lipid degradation in the crucial steps of embryogenesis is novel.

The experiment in which chimeric embryos composed of both CIDEA KO and WT wells exhibit disorganization of the epiblast and generation of multiple lumens, demonstrating the crucial role of intracellular lipid storage and CIDEA protein in the mammalian embryos is relevant and original.

The idea that the mobilization of the lipids into the lysosomes could be a crucial process for the cellular polarity of the epiblast cells and in apical lumen formation is very novel.

In the light of their data it would be worthy to extend slightly the discussion on LDs utilization to preimplantation blastocyst, particularly including blastocyst growth during delayed implantation. The same suggestion also refers to LDs metabolism and cellular signaling in the blastocyst. The authors mention about both aspects just in one sentence.

REVIEWER COMMENTS _ NCOMMS-21-13681-T

Reviewer #1 (Remarks to the Author):

This manuscript from the Azura lab explores the role of the lipid droplets previously observed in preimplantation and diapause embryos, which are subsequently dispersed after implantation. The authors show that LD size increases during preimplantation development, accompanied by decrease in number, implying LD fusion. They propose that this process is a prerequisite for the morphogenic change in epiblast from apolar to epithelial. CIDEA protein was found to be much more abundant in ESCs compared with MEFs in a screen for likely candidates. OE of Cidea in ESCs caused increase in LD size and decrease in number, whereas KO reduced size and increased number, consistent with its role in LD fusion. The authors further show, using ESCs, that multiple lumens form in 3D aggregates of Cidea KO cells, whereas only one central aggregate formed in WT or Cidea-WT rescue aggregates. WT versus KO ESCs were injected into host embryos and dissected at E5.5. Contribution of KO cells over a threshold level caused disorganisation of the epithelium. Treatment of KO spheroids with DEUP (cytosolic lipase inhibitor) could rescue LDs and (partially) central lumen formation. By testing appropriate inhibitors, the mode of operation for LD lipid release in lumen formation was attributed to lipophagy. Chimaeras generated from in Cidea KO ESCs show multiple lumen formation at E5.5, consistent with the 3D in vitro Cidea KO ESC cultures. However, by E7.5 only one lumen is observed, but whether the embryo is normal is not clear from the poorly presented images in Supp. Fig. 6g. LDs are observed in the VE, which the authors propose may contribute to the rescue of the Cidea KO chimaeras. There must be some other compensatory mechanisms, since Cidea KO in mice does not result in an embryonic lethal phenotype. The methods are very well documented, and the statistics seem appropriate. This study is quite interesting, but a few improvements need to be made to the manuscript:

We thank the reviewer for their positive evaluation of our study. We have now improved several of our figures and provide more discussion of our findings, as outlined in our responses to the reviewers.

1. Fig. 2h: the E3.5 image chosen may not be the best example, since only one of the LDs shows CIDEA protein clearly. Also, it is not clear from the image to which cell this LD belongs. It is very close to the CDX2-positive nucleus, therefore more likely to be TE than ICM. Inclusion of a membrane marker, such as the E-CAD used in Supp. Fig. 4c, would address this question.

We now present an alternative image illustrating the accumulation of CIDEA protein at the surface of enlarged lipid droplets (LDs) in E3.5 blastocysts (see New Figure 2h). Here, phalloidin was used as a membrane marker, and the identity of trophectoderm (TE) and inner cell mass (ICM) cells was assigned based on the location of YAP protein, which is selectively excluded from the nucleus of ICM cells at this developmental stage (Hirate *et al.*, 2012; Nishioka *et al.*, 2009). A total of 25 embryos were stained across 4 experiments; out of those embryos, 21 of them showed at least one clear example of CIDEA protein staining along BODIPY-labelled LDs.

Please note that the anti-CIDEA antibody (sc-8730-R, Santa Cruz), which we previously used, has been discontinued and therefore we tested for this revision work several new antibodies with satisfying results obtained with an anti-CIDEA antibody purchased from Proteintec (13170-1-AP). To acknowledge that CIDEA protein can be detected along LDs using two independent antibodies *in vivo*, our previous staining example is also presented in Supplementary Figure 1h.

2. The qRT-PCRs for pluripotency and differentiation factors presented in Supp. Fig. 2 should be brought into the main figures, since they provide important evidence that the disruption to luminogenesis caused by Cidea KO does not impede exit from naïve pluripotency.

The RT-qPCRs presented in Supplementary Figure 2 do not provide direct evidence that the disruption to lumenogenesis caused by CIDEA KO and the exit from naive pluripotency are uncoupled events upon differentiation (3D spheroid cultures and removal of LIF) since these data were generated from ESCs grown under self-renewing conditions (serum/LIF). We have now clarified the description of Supplementary Figure 2g-l in our revised manuscript page 6, lines 126-130.

To specifically address the point of the reviewer, we have generated new RT-qPCR data in 3D spheroids formed by WT and CIDEA KO ESCs at successive differentiation time-points (0h, 24h, 48h and 72h), validating that the exit from naïve pluripotency operates regardless of CIDEA expression as evidenced by the downregulation of *Esrrb* and upregulation of *Otx2* at similar rate in both cultures. These new data are presented in the main Figure 3b and complements our global RNA-sequencing analysis shown in Supplementary Figure 5 (see revised manuscript page 7, lines 171-173).

3. In Supp. Fig. 4c the mural TE contains multiple large LDs, but there are apparently none in the polar TE. Could the authors speculate on the destination of these, since it is the polar TE that is required for differentiation of the extraembryonic ectoderm and subsequently the chorion?

The asymmetry between polar and mural TEs in terms of large LD content is indeed very intriguing and is now further highlighted in our revised manuscript (see pages 6-7, lines 150-152). This may provide a novel marker to delineate the tissue boundary between the polar and mural parts of the TE when the blastocyst initiates its restructuring into an egg cylinder.

In terms of the purpose of LDs in polar TE cells and given that LDs show similar dynamics of enlargement and mobilization in epiblast and polar TE cells, it may be tempting to speculate a conserved role for LDs in orchestrating the morphogenesis of embryonic and extra-embryonic tissues upon implantation; a view that we now discussed with a note of caution in our revised manuscript pages 12-13, lines 298-304.

4. The chimaera images in Supp. Fig. 6g should be replaced with images in a more conventional orientation with IHC for a couple of lineage markers to aid navigation for the reader.

We have now improved the presentation/orientation of E7.5 embryo chimaera images shown in Supplementary Figure 6i to aid navigation for the reader. The rescued formation of the pro-amniotic cavity in CIDEA KO chimaeras is not surprising given the non-embryonic lethality of CIDEA KO mice (Zhou *et al.*, 2003) and indicates the existence of compensatory mechanisms independent of CIDEA expression as outlined in our revised manuscript page 9, lines 208-211 (see also discussion pages 13-14, 319-332).

LDs are observed in the VE, which the authors propose may contribute to the rescue of the Cidea KO chimaeras. There must be some other compensatory mechanisms, since Cidea KO in mice does not result in an embryonic lethal phenotype.

Based on the above reviewer's feedback, we removed from our discussion the view that the visceral endoderm (VE), which continues to accumulate larger LDs in the post-implantation embryos, could supply lipids to the epiblast and thus contribute to the rescue of CIDEA KO chimaeras. Instead, we focus our discussion on mechanisms that may operate in both CIDEA KO mice and embryo chimaeras including a potential switch from polarity-driven to apoptosis-driven mechanisms of cavity formation (see pages 13-14, lines 319-332).

Reviewer #2 (Remarks to the Author):

In this manuscript, Hang T et al. revealed the dynamic changes of lipid droplets (LDs) and their importance during post-implantation embryogenesis. The authors found that LDs expand after the morula stage due to CIDEA protein-mediated LD fusion, and that lipophagy is involved in the degradation of the enlarged LDs. Using an original spheroid model, they also showed that dysfunction of CIDEA protein and suppression of lipophagy affect lumenogenesis, demonstrating that lipid storage plays an essential role in peri-implantation embryonic development. In general, the evidence in this study is sufficient to support the authors' conclusions, and these results are potentially interesting. However, there are several aspects that need to be resolved/improved before this manuscript can be considered for publication in Nature Communications.

We thank the reviewer for highlighting the scientific soundness of our study and for their insightful and positive comments.

Major points:

1. One of my concerns is that most of the results rely on fluorescent microscopy imaging of LDs. For example, when LD synthesis or degradation is inhibited or autophagic activity is blocked, the author should examine not only changes in LD distribution by BODIPY staining, but also changes in intracellular LD content. For this, Western blotting with antibodies against LD-specific proteins such as Plin2 protein and measurement of intracellular triglyceride content may be effective. This analysis should also be done in experiments with ATG5- and CIDEA-deficient/mutant spheroids.

As suggested by the reviewer, we have now confirmed our quantifications of neutral lipid content based on total BODIPY intensity by direct measurements of intracellular triglyceride levels in differentiating 3D spheroids formed by WT, CIDEA KO and ATG5 KO ESCs using fluorometric assays (see New Supplementary Figure 5a and New Supplementary Figure 8f). To avoid any confusion between the two assays, we have also changed the axis labelling of Figure 3d and Supplementary Figure 8e for "BODIPY Intensity" instead of "Neutral lipid content" as well as throughout our revised text.

We also ran Western Blotting with anti-PLIN2 antibodies on total protein extracts collected from differentiating WT, CIDEA KO and ATG5 KO spheroids. However, the results were inconclusive and therefore are not included in our revised manuscript. This is mainly because the *Plin2* transcript itself is developmentally downregulated upon 3D spheroid differentiation (our unpublished observation), making difficult to ascertain whether PLIN2 protein loss is caused by its transcriptional silencing or targeted protein degradation.

However, and as highlighted in our revised discussion (see page 13, lines 311-318), we think that determining in the future the full LD-associated proteomes of WT and mutant 3D spheroids upon differentiation could be more informative, potentially shedding light on how lipid storage and its timely mobilization are regulated in pluripotent cells at a key developmental transition.

2. Another point of my concern is the localization of CIDEA protein to the LD surface. In general, it is known that CIDEA protein are enriched at the contact site between LDs to facilitate LD enlargement (as shown in Fig. S3C). However, since the localization of CIDEA protein shown in current manuscript is mostly observed uniformly on the LD surface, the authors need to confirm whether CIDEA protein is enriched at the LD contact sites under the conditions and/or stages when LDs fuse and grow larger.

As suggested by the reviewer, we have now validated that the V5-CIDEA protein as revealed by either anti-V5 or anti-CIDEA (13170-1-AP; Proteintech) staining is enriched at LD-LD contact sites in ESCs upon induction of LD fusion and enlargement (i.e., upon short treatment with KSR; ~ 6-7 hours) - please see New Supplementary Figure 2c and revised manuscript page 5, lines 118-121).

We note, however, that the accumulation of CIDEA protein on the surface of enlarged LDs is somehow enhanced and/or more uniform when ESCs are cultured under KSR conditions for longer period of times (i.e., 24-48 hours) (see Figure 2d,g and Supplementary Figure 3d), suggesting that CIDEA protein expression could be stabilised upon exposure to fatty acids in excess via an unknown mechanism.

3. I agree with the authors' view that autophagy (lipophagy) is involved in the degradation of enlarged LDs because the suppression or loss of autophagy results in the accumulation of enlarged LDs. However, the current results are not sufficient to clarify the lipophagy-mediated degradation of the enlarged LDs. The authors claimed the involvement of lipophagy based on the finding that lysosomes are in close proximity to enlarged LDs, but it is also possible that lysosomes directly fuse with LDs. Therefore, in order to further clarify the involvement of lipophagy, it is necessary to 1) observe the localization of autophagosomes in enlarged LDs, and 2) investigate the changes in LD content under conditions of autophagy suppression or deficiency (the second point is related to comment 1).

To further clarify the involvement of lipophagy, we performed additional dual staining for LC3 (a marker of autophagosomes) and BODIPY on 3D spheroids formed by WT ESCs (New Supplementary Figure 8a) showing that LC3 is recruited to the surface of enlarged LDs from 24 hours post induction of differentiation. Similar staining was also performed on early and late blastocysts (E3.5 and E4.5) *in vivo* and the co-localization of LC3 (autophagosomes) and BODIPY (LDs) signals quantified (New Supplementary Figure 8b,c) as previously done for lysosomes and LDs (Figure 5d,e). Collectively, our results demonstrate an increased incidence in the co-localization of LDs with LC3 (new data) and lysosomes (previous data) at the onset of LD mobilization both *in vitro* and *in vivo*. This indicates the involvement of autophagosomes, trapping LDs, prior to their fusion with lysosomes as now described in our revised manuscript page 11, lines 260-266.

Further supporting this conclusion, we also confirmed by direct measurements of LD-stored triglycerides that the dynamic changes in LD content we observe in differentiating WT 3D spheroids is disrupted upon ATG5 depletion and inhibition of autophagic fluxes (New Supplementary Figure 8d-f and revised manuscript page 11, lines 269-270). Please see also page 14, lines 339-347 for further discussion on autophagic fluxes in 3D spheroids formed by WT, ATG5 KO and CIDEA KO ESCs, as examined by LC3 western blotting during this revision work.

4. In Fig.S6d, the authors should not only examine the localization of LDs to the Golgi apparatus in the spheroid once expanded LDs are observed, but also to mitochondria and lysosomes if LD mobilization is active at that time.

As suggested by the reviewer, we have now performed additional staining for LDs (BODIPY) along with mitochondria (via ATPB; New Supplementary Figure 6e) and lysosomes (via LysoTracker; New Supplementary Figure 6f) at the time of LD mobilization in both WT and CIDEA KO 3D spheroids. These new results confirm that

- 1- All intracellular organelles examined - including the Golgi apparatus, LDs, mitochondria, and lysosomes, are preferentially repositioned at the sub-apical domains of polarised WT spheroids. In contrast, their distribution tends to be more

scattered/disrupted in CIDEA KO spheroids (48 hours; new Supplementary Figure 6e,f) (see also revised manuscript page 8, lines 185-189).

- 2- Mitochondria-LD interactions are relatively infrequent when compared to lysosome-LD interactions in differentiating WT 3D spheroids at the time of LD mobilization (see page 11, lines 255-257). This resonates with our unpublished observations that inhibiting fatty acid oxidation (via Etomoxir treatment) does not impede the morphogenesis of 3D spheroids in contrast to autophagy suppression.

Minor points:

5. The authors mention "release" of lipids in the title, abstract, and text, but their study does not experimentally demonstrate uptake or release of lipids into LDs, so please reconsider the title and use of "release" in this regard. LD enlargement (via CIDEA protein) and their degradation (by lipophagy) is an important finding, so why not include this in the title?

The title of our manuscript has been changed. It now reads as "Dynamic Enlargement and Mobilization of Lipid Droplets in Pluripotent Cells Coordinate Morphogenesis during Mouse Peri-implantation Development".

We have also substituted the word "release" by either "mobilization" or "degradation" throughout the manuscript.

6. Throughout the manuscript: the developmental time of the fertilized embryos up to the blastocyst stage should not only be indicated by embryonic day, but should also be combined with the time after hCG, for example, to better define the developmental stage. This is because the morphology of LDs seems to change rapidly before and after compaction. Related to this, the exact time (stage) of fertilized embryos indicated as "pre-compaction" should be specified (Figure 1a). Is it 4-cell stage?

This information has been added in the legends of Figure 1a, Figure 5d,e and Supplementary Figure 8b,c.

As for Figure 1a itself, 4-8 cell embryos were harvested ~ 68 hours post hCG injection, cultured for 6 hours to E2.5 (morula; ~74 hours post hCG injection) and for another 24 hours to E3.5 (blastocyst; ~98 hours post hCG injection) as now clarified in our revised manuscript page 4, lines 89-91.

7. Currently, it is unclear whether the amount of intracellular LD content actually increases ("storage") or decreases ("mobilization") during spheroid formation, simply based on the morphological changes of LDs by BODIPY staining (as pointed out in comment 1). Depending on the outcome of comment 1, the overview figure in Figs. 5i and S7a should be revised (if necessary) and a more detailed explanation should be provided in the text.

Our new measurements of triglyceride levels confirm that the amount of LD-stored lipids dynamically increases ("storage") and decreases ("mobilization") in WT spheroids by 24 hours and 48 hours post-induction of differentiation respectively as previously shown by quantifying total BODIPY intensity (see also major point 1).

8. P4, line 79: Since the last paragraph contains the results of CIDEA (lines 100-108), please reconsider the subtitle? Alternatively, the paragraph that mentions about CIDEA data could be moved to the next section.

Although we have chosen to not change the subtitle of this section, we edited the text (see page 5, lines 109-116) to explicitly mention other factors involved in LD growth and enlargement, thus putting less stress on the *Cidea* transcript at this point of our manuscript.

9. Some of the enlarged images (Figs. 2g and 2h, for example) are almost the same size as the original images, so they should be changed to more enlarged image.

This has now been checked throughout our figures and adjusted where appropriate.

Along the same lines, CIDEA protein in blastocysts appear to be partially accumulated rather than localized in a "ring" (page 6, line 129) on the LD surface (Fig. 2h). It might be clearer if higher resolution images could be taken.

The mention of CIDEA protein "ring" has been deleted in our revised manuscript (page 6, line 139-142). Instead, we now refer to the accumulation of CIDEA protein along the surface of enlarged LDs.

In addition, in some figures (Figs. 1a, 2a, 2d, 2g, 3h, and 5d), the area showing the enlargement and the area of the actual enlarged image are slightly misaligned.

We have checked and amended where appropriate these figures to ensure correct alignment of area delineated by dotted boxes with the area being enlarged in zoomed views.

10. Typos: Page 14, line 335: "Narashige" should be Narishige.

This has been corrected in our revised manuscript.

Reviewer #3 (Remarks to the Author):

I found this article very interesting. The authors used a range of methods for the study of CIDEA protein in LDs utilization by embryos. It is worthy to highlight that the authors verified their hypothesis on multiple models, analyzing the lipid droplet size formation in the blastocysts, stem cells derived spheroids and finally in the chimeric embryos. To study the role of CIDEA protein, they generated the CIDEA knockout stem cell line (created with the use of CRISPR-Cas 9 methodology) that served for both to study the CIDEA function in directly in the embryonic stem cells and during the formation of spheroids and chimeras. Moreover authors used the fluorescence and immunofluorescence to study the CIDEA and lipid colocalization and the real time PCR to study the expression of the key genes involved in the lipid dynamics control. With the use of specific chemical inhibitors authors confirmed the role of the lipophagy process in the embryonic LD utilization (as opposed to lipolysis that seems to be irrelevant in the embryonic lipid droplets utilization). The finding that CIDEA protein is a potentially key regulator of the LD fusion in embryonic stem cells and in the same time regulates the lipid degradation in the crucial steps of embryogenesis is novel. The experiment in which chimeric embryos composed of both CIDEA KO and WT wells exhibit disorganization of the epiblast and generation of multiple lumens, demonstrating the crucial role of intracellular lipid storage and CIDEA protein in the mammalian embryos is relevant and original. The idea that the mobilization of the lipids into the lysosomes could be a crucial process for the cellular polarity of the epiblast cells and in apical lumen formation is very novel.

We thank the reviewer for their enthusiasm highlighting the novelty of our findings.

In the light of their data it would be worthy to extend slightly the discussion on LDs utilization to preimplantation blastocyst, particularly including blastocyst growth during delayed implantation.

The same suggestion also refers to LDs metabolism and cellular signaling in the blastocyst. The authors mention about both aspects just in one sentence.

We have extended our discussion of current knowledge pertinent to LD building-up and utilization in mouse pre-implantation embryos. Notably, we provide more contextual insight into two recent studies (Tatsumi *et al.*, 2018; Aizawa *et al.*, 2019), which elegantly addressed the importance of maintaining the correct number of LDs for the proper formation of the blastocyst (see page 12, lines 279-291).

As suggested, we are also discussing our findings in relation to recent reports showing that blastocysts in diapause mobilize LDs, most likely fuelling fatty acid oxidation to support the survival of developmentally arrested embryos (Arena *et al.*, 2021). In agreement, it has also been reported that diapause is associated with lipolysis-mediated degradation of LDs upon AMPK activation and mTORC2 suppression (Abdiasis *et al.*, 2019), also signposting the importance of these two signalling pathways in the blastocyst at peri-implantation times (see page 15, lines 348-361).

REVIEWERS' COMMENTS

Reviewer #2 (Remarks to the Author):

The authors thoroughly revised the manuscript to address my questions and concerns. Importantly, they added many new experiments and performed extensive rewrites to improve the manuscript. Therefore, I recommend publication of the paper after these revisions. However, there are a few minor corrections and comments:

1. Narashige should be corrected to Narishige (page 17, line 401).
2. Fig.S8d: It would be better if typical blot data were shown in the figure (if possible).

REVIEWERS' COMMENTS_ NCOMMS-21-13681A

Reviewer #2 (Remarks to the Author):

The authors thoroughly revised the manuscript to address my questions and concerns. Importantly, they added many new experiments and performed extensive rewrites to improve the manuscript. Therefore, I recommend publication of the paper after these revisions. However, there are a few minor corrections and comments:

1. Narashige should be corrected to Narishige (page 17, line 401).

Thank you very much for thoroughly going through our revised manuscript. This has now been corrected.

2. Fig.S8d: It would be better if typical blot data were shown in the figure (if possible).

The same bar plot data representation (quantification of Western blot bands and ratio analysis) has been kept now with all point data shown as symbols in the figure and exact p-values provided in legend. Please also note that all associated raw data including uncropped blots and quantification values are provided as a Source Data file with this study.